# "Give me a break!" A systematic review and meta-analysis on the efficacy of micro-breaks for increasing well-being and performance

**Patricia Albulescu**, **Irina Macsinga***, **Andrei Rusu, Coralia Sulea, Alexandra Bodnaru, Bogdan Tudor Tulbure**

Department of Psychology, West University of Timioara, Timioara, Romania

* irina.macsinga@e-uvt.ro

## Abstract

Recovery activities during short breaks taken between work tasks are solutions for preventing the impairing effects of accumulated strain. No wonder then that a growing body of scientific literature from various perspectives emerged on this topic. The present meta-analysis is aimed at estimating the efficacy of micro-breaks in enhancing well-being (vigor and fatigue) and performance, as well as in which conditions and for whom are the micro-breaks most effective. We searched the existent literature on this topic and aggregated the existing data from experimental and quasi-experimental studies. The systematic search revealed 19 records, which resulted in 22 independent study samples ($N$ = 2335). Random-effects meta-analyses shown statistically significant but small effects of micro-breaks in boosting vigor ($d$ = .36, $p$ < .001; $k$ = 9, $n$ = 913), reducing fatigue ($d$ = .35, $p$ < .001; $k$ = 9, $n$ = 803), and a non-significant effect on increasing overall performance ($d$ = .16, $p$ = .116; $k$ = 15, $n$ = 1132). Sub-groups analyses on performance types revealed significant effects only for tasks with less cognitive demands. A meta-regression showed that the longer the break, the greater the boost was on performance. Overall, the data support the role of micro-breaks for well-being, while for performance, recovering from highly depleting tasks may need more than 10-minute breaks. Therefore, future studies should focus on this issue.

## Introduction

In an "always-on" culture encouraged by the Fourth Industrial Revolution [1], it is essential to find a balance between being effective at work and having optimal well-being. Recent reports highlight the "human energy crisis" many employees face today [2, 3]. Heavy workloads and long hours impede their capacity and energy renewal [4]. Accordingly, scholars from different areas, from organizational psychology or ergonomics to experimental psychology, have been paying attention to mechanisms related to recovery from effort in both employees and students [5]. Therefore, a growing body of literature focuses on momentary recovery and energy management strategies during working time [6, 7].

Energy, as well as effort, is required in achieving work-related tasks and objectives. Work demands can deplete psychological resources [8], having a strong correlation with exhaustion

**Data Availability Statement:** All relevant data are within the article and its Supporting Information files.

**Funding:** The work of A. R. was supported by a grant of the Romanian Ministry of Education and Research, CNCS - UEFISCDI, project number PN-III-P1-1.1-TE-2019-2032, within PNCDI III.

**Competing interests:** The authors have declared that no competing interests exist.

[9] and fatigue [10]. After expending energy over a while, a process of recovery or replenishment is needed [11]. Individuals have several possibilities to recover and build new resources, and during more extended periods of free time, such as evenings [12], weekends [13], or vacations and sabbaticals [14]. Importantly, recovery happens also at shorter intervals during formal working hours, such as lunch breaks [15], scheduled breaks [16], or micro-breaks [17].

The concept of micro-breaks originates in the ergonomics literature, defined as scheduled rests that individuals take to prevent the onset or progression of physical symptoms, such as musculoskeletal pain or discomfort [18]. In the organizational literature, this concept was introduced as a brief resource-replenishing strategy, taken informally between work tasks [19, 20]. Besides micro-breaks [21], several other terms are widely used to refer to short internal recovery, such as work breaks [22], rest breaks [16], energy management strategies [5], recovery behaviors [23], restorative activities [24], and mini-breaks [25]. Micro-breaks are beneficial for the worker's well-being and job performance [19, 26], even if the total work time is reduced because of the breaks [16]. For the purpose of this paper, we adopted a general definition of micro-breaks as short discontinuities in one's tasks of no longer than 10 minutes [17, 27]. Although a consensus was not reached on the optimal duration of a micro-break, or even on how short a short break is, thus creating a fairly high variability in time-on-break between studies, we rely both on the categorization of recovery time by Sluiter et al. [27], and the qualitative exploration of Bennett et al. [17]. Specifically, Sluiter et al. [27] used the term "microrecovery" to define what happens in the first minutes after a period of effort exertion, considering short pauses between tasks. Because the next category of recovery time includes a period between 10 minutes and about 1 hour after engaging in work-related tasks, we identified the 10-minutes limit as the maximum amount of time allocated for micro-breaks. Moreover, Bennett et al. [17] coming across the same issue related to micro-break duration, conducted a series of semi-structured interviews supporting the time-cap of 10 minutes for such breaks. Thus, we selected the term micro-breaks as it best defines the duration (micro; 10 minutes or shorter) and the action (break).

A growing body of literature focused on the recovery after the work-related energies have been exhausted [28–30]. However, the process of recovery that happens during the workday or between work tasks is still insufficiently analyzed, and conclusions on their effects are still not clear. The present meta-analysis addresses these limitations, focusing on the recovery process by including experimental studies investigating the momentary impact of micro-breaks between work tasks on well-being and performance.

## Theoretical underpinnings and outcomes of micro-breaks

We can rely on multiple explanatory theories to understand how micro-breaks act on individual outcomes. From a fundamental standpoint, the cognitive load theory states that the mental capacity in working memory is limited, and if a task requires too much ability, learning will be hindered [31–33]. Individuals have limited cognitive resources; when allocating resources to one task, their availability becomes limited for other jobs [34]. In this explanatory context, micro-breaks can be seen as natural reactions of the cognitive system to a possible cognitive overload that could affect performance.

Narrowing down towards the applied workplace context, the Conservation of Resources theory (COR) [35] and the Effort Recovery Model (ERM) [8] provide the theoretical foundation for the role of recovery in the relationship between work demands, resources, and stress. The general assumption is that employees have a particular supply of personal resources, such as directed attention or mental resilience instrumental for achieving work goals [36]. Recovery is thus necessary, achievable when no demands similar to those related to the task at hand are

put on the person [8], or when new resources are built up (e.g., energy, feelings control) [35]. Individuals lacking recovery experiences tend to endure fatigue and to feel negative affect [37], whereas recovered individuals feel more vigorous and engage in helping behaviors [38]. More-over, the restoration theories such as Attention Restoration Theory (ART) [39] and The Stress Recovery Theory (SRT) [40] are traditionally used to explain the mechanisms through which exposure to nature can improve mental well-being and performance by reducing the impact of stress, mental fatigue, and negative affect [41–43].

In terms of specific outcomes, there are (at least) two individual-level components of well-being relevant for recovery: vigor (a pleasant activation) and fatigue (unpleasant deactivation) [28, 44]. Moreover, in COR theory, energy is an intrinsic resource (i.e., vigor) that must be replenished when exhausted [45]. Vigor contributes to the willingness to invest effort into the tasks at hand and persist when difficulties arise [46]. For instance, in one study many of employees reported break activities were negatively associated with increased energy (i.e., vitality) but positively related to fatigue [5]. Such results suggest that "employees seek out these strategies when they are already fatigued" [5, p. 34]. The results of a diary approach suggest that break activities positively predict vigor but negatively predict fatigue [21].

Performance represents another key outcome on which micro-breaks are considered to have an impact. It is well known that cognitive (i.e., declarative knowledge, procedural knowl-edge, and skills) and motivational factors (i.e., effort investment and persistence) are the main determinants of work performance [47]. Breaks from work can improve task performance through beneficial resource-strain, cognitive, affective, and motivational mechanisms [16]. Breaks are essential for performance on sustained attention tasks, suggesting that the vigilance sensitivity decrement is influenced by the frequent use of cognitive resources [48].

However, the current knowledge on micro-breaks relies considerably on experimental data testing their effects, such as improved self-control capacity, mood, or work engagement [49–51]. Specific answers are still needed to understand the effects of micro-breaks on well-being and performance, to explore whether they have an optimal duration and how vital the contex-tual factors are. To our knowledge, these important questions have no clear answers to date.

Based on the theoretical and empirical rationale described above, we explored two main questions in this meta-analysis. The first refers to the efficacy of micro-breaks (as defined ear-lier) on participants' well-being. In other words, we would like to see if we could collect enough meta-analytical evidence to support the assumption that micro-breaks increase indi-viduals' vigor and decrease their fatigue levels. The second research question refers to the effi-cacy of micro-breaks in enhancing participants' performance. In this respect, we would like to see whether we could find sufficient meta-analytic support for the idea that, despite slightly reducing the time allocated to the task, could micro-breaks actually augment one's perfor-mance? In addition to these central questions, we wanted to refine our finding by exploring the impact of several moderator variables.

## Potential moderators for the effectiveness of micro-breaks

**Break activity and duration.** Breaks lead to recovery when individuals engage in activities that reduce the demands put on their resources [5, 52]. During work hours, recovery activities can be related to task objectives (e.g., helping a colleague; setting up a new work-related goal) [5, 53, 54], or can be unrelated to the job (e.g., attending to physiological needs; engaging in social interactions; cognitive; relaxing; directing attention to natural elements) [5, 21].

Overall, work-related micro-break activities were associated with decreased well-being, decreased sleep quality, and increased negative mood [55]. Physical activities such as stretching and exercise were associated with increased positive emotions and decreased fatigue [56, 57].

Relational activities (e.g., checking in with friends and family members) were associated with increased feelings of vitality [21]. The use of personal social media and games was associated with less conflict between work and private life [58], whereas watching a short movie clip was associated with increased recovery and performance [59].

Regarding the duration of micro-breaks and their impact on well-being and performance, some studies suggested that recovery effects could be elicited after a very short time (i.e., 27.4 seconds) [60]. Another study showed that 40-s micro-breaks are sufficient to improve attention and task performance [61]. Finally, other scholars were more generous with the time needed to recover during micro-breaks, from a few seconds to several minutes, implying the possibility that micro-breaks may have an "optimal duration" [20]. However, there is still no established standard regarding the length of such short breaks, as well as no explicit consideration on how much time is sufficient for recovery to occur [17].

Considering the significant difference in the proposed duration of micro-breaks and the fact that there is "little hard evidence concerning the optimum length of rest breaks" [62, p. 123], we would also like to address the following question: Does the efficacy of micro-breaks differ as a function of break activity and break duration?

**Study design, setting, sample characteristics, and contextual factors.** Because resource expenditure and recovery can be affected by several contextual and personal factors [63, 64], we consider several additional moderators.

In relation to the type of task from which the participants are recovering, research in neuroscience and cognitive science highlights the relationship between mental effort, complex cognitive tasks, and working memory capacity [65–68]. The more complex a task is, the more mental effort is required. This effort increases neural activity and the metabolic demands on the brain [69]. As a result, fatigue sets in quickly, working memory becomes overloaded, and the recovery effort will be more significant [31]. Therefore, the type of cognitive task the individual is involved in before the break becomes relevant in studying the efficacy of micro-breaks on well-being and performance.

The study setting for the experiment (e.g., laboratory vs. field) is also of interest as a moderating factor, because usually there are no direct personal consequences of one's behavior in the case of participants in a laboratory experiment, where controlled breaks are taken as instructed. In contrast, participants within the working schedule may experience perceived pressure from colleagues or superiors to keep working to avoid looking "lazy" or uninvolved with the organizational objectives.

When researchers test the effect of micro-breaks on individual outcomes, they tend to use convenience samples of students [17]. Hence, the category of participants (i.e., students vs. employees) may also translate into different effects of the experimental manipulations since the populations may differ in their fundamental motivation to participate in such studies.

Another aspect that could moderate the effect of the break on the cognitive resources' restoration is the type of break taken by the control group (i.e., no break vs. some form of task interruption) that is generally used in experimental studies. Therefore, in experimental designs where the control group had no breaks, we expect the effect of breaks for the intervention group to be more substantial simply because the participants in the control group did not interrupt their work and had no time for energy recovery [8].

Finally, the measurement and conceptualization issues are primarily relevant for the performance outcome because performance can mean different things in different environments [70]. For some work positions, performance could mean having a prompt reaction time, for others could mean displaying correct responses, while for others could mean generating a new and divergent set of ideas. Moreover, self-reported performance represents a subjective perception that is susceptible to distortions. When the effect of breaks on performance is

experimentally tested, it is typically anchored in an active sustained attention research approach (measured by means of highly demanding speed tests) or in a more passive sustained attention approach (measured by means of detection tasks), only recently being used a simpler, sustained-attention, reaction-timed task, requiring speed responses to simple targets (measured with psychomotor vigilance test) [70]. Therefore, our final question investigates the effect of such methodological factors. As far as the data allowed us, we wanted to delineate the impact of a) the task performed before taking the break (e.g., cognitive or creative activities); b) the setting of the study (e.g., workplace vs. laboratory); c) the professional category of participants (e.g., employees vs. students); d) the type of control group (e.g., also in break vs. no break), and e) the operationalization of the performance outcome. Consequently, we explored to what degree the efficacy of micro-breaks differs between various contextual factors (i.e., the design characteristics detailed above).

## Method

We used the PRISMA 2020 framework to conduct and report the systematic review and meta-analysis [71]. The protocol was registered on the PROSPERO platform (ID: CRD42021242961).

### Eligibility criteria

Based on the PICOS approach, the included studies had to meet the following criteria: the studied population to be represented by healthy individuals, either employees or students involved in depleting work or comparable work-related tasks (to be able to generalize the findings to the workplace context) (P); to be embedded in micro-breaks literature or focused on short break activities during tasks of 10 minutes maximum (I); a control group to be included as a comparison (C); among the monitored outcomes to be at least one of interest, such as vigor, fatigue, or performance (O); and to have a between-groups experimental or quasi-experimental design (S). Also, the studies had to be published in the last 30 years, written in English (language), and available in full text (availability). The 30 years for this search was used according to Scholz et al.'s [22] argument on the changes in the workplace due to the inclusion of computers and subsequent changes following the reliance on technology.

Moreover, the studies were eligible if they included all the necessary data in computing effect sizes. If these details were not reported, the corresponding author was contacted to provide them.

### Information sources and search strategy

The systematic search strategy was based on enquiring exclusively online databases (SCOPUS, MEDLINE, and PsycINFO) for research published between January 1990 and April 2021. Searches were performed using Boolean operators. Details about how the search was conducted for each database can be found in S1 Table.

### Selection process and data management

We used a two-step approach in study selection: (1) titles and abstracts screening and (2) full-text screening of the remaining records. In the first step, we enquired if the studies investigated internal rest breaks and the outcomes of interest, such as well-being and performance, the design of the studies, and details about the population tested. The next step in the study selection process consisted of screening the full texts of records selected in the previous step and

applying the eligibility mentioned above criteria. Cited references were also checked, and eligibility criteria were also applied to these records.

Two independent reviewers (PA and AB) did the screening of both titles and abstracts, and full-text records was done by two independent reviewers (PA and AB). Any disagreement between them over the eligibility of particular studies was resolved through discussions with a third reviewer (AR).

## Data items and collection process

For each eligible study, we proceeded to extract the following information: general bibliographical information (i.e., authors and year of publication); sample characteristics such as country, professional category (i.e., employees or students), age, and gender (P); recovery activity duration (expressed in minutes), type (i.e., work vs. non-work-related; and cognitive, physical, relational, relaxing, or nature-related), and setting (i.e., organization vs. laboratory) (I); type of control group (i.e., also break vs. no break) (C); type of outcome (i.e., vigor, fatigue, performance) and the operationalization (O); research design (i.e., randomized vs. non-randomized) (S). Moreover, we included antecedents of recovery as the type of work task (i.e., clerical, cognitive, creative, or emotional work), the relevance of the task for the workplace (relevant or irrelevant), as well as task duration (i.e., time on the task before the break, expressed in minutes).

We recorded effect sizes for measures of vigor, fatigue, and performance (objective and subjective). Objective performance is expressed as a measure of accuracy representing the amount of work performed in a specific time (e.g., mean correct responses, number of words remembered, errors reported during the completion of tasks, etc.), response speed (reaction time), as well as ideas generated (e.g., where the task require generating new ideas on a topic). Subjective performance variables were measures of self-reported levels of work productivity. Vigor and fatigue represent scores on subjective, self-report instruments.

The data collection and coding process were performed by two reviewers working independently (PA and AB). Differences or inconsistencies were re-examined with a third reviewer (AR) until an agreement was reached. To evaluate the degree of agreement between the reviewers, we computed Cohen's kappa coefficient. The results suggest a moderate average agreement between the assessors (72.56%; average k = 0.45).

## Risk of bias

The internal validity of the studies was performed based on the Cochrane risk of bias tool [72] by taking into consideration six domains: sequence generation and allocation concealment (selection bias), blinding of outcome assessor (detection bias), incomplete outcome data (attrition bias), selective outcome reporting (reporting bias), and other potential threats to internal validity.

Each criterion was evaluated for each study by assigning it a "low risk," "high risk," or "unclear risk" of bias rating. At the low risk of bias were labelled the studies which reported a clear description of how that specific internal validity criterion was handled. The more criteria with a low risk of bias a study meets, the higher the study's internal validity can be concluded.

The same two assessors performed the risk of bias assessment. The same third expert discussed any incongruence between them and consensually settled. Inter-rater agreement was consistent across domains, ranging from fair ($k = 0.27$ for blinding of outcome assessor and participants) to substantial ($k = 0.67$ for incomplete outcome data).

## Summary measures and synthesis of results

The main research questions were addressed using random-effects meta-analyses based on Borenstein et al.'s [73] framework. We used effect size estimates based on the between-groups standardized mean difference (Cohen's d) [74], with 95% confidence intervals and two-sided p-values. Positive estimates represent effects in the hypothesized direction (i.e., increased vigor and performance; decreased fatigue), while negative values indicate opposed ones.

Because only two studies collected and analyzed follow-up data [75, 76], we decided to include only the post-manipulation data in our analysis.

To ensure that all effect sizes are computed on independent groups, in cases where studies assessed the effect of different micro-break interventions against the same control group, we combined those intervention groups into one by using standardized or pooled data [17, 49, 59, 75, 77–80]. For the study conducted by Rees et al. [81] several steps were taken, namely: (1) for each of the five post-interruption blocks for each outcome variable tested (i.e., correct responses, reaction times, response bias, sensitivity), the average was calculated; (2) from every five experimental groups (i.e., free break, music, music with video, choosing between listening to music or watching a music video, no activity break), because they were tested against the same control group, we created a single intervention group by calculating their weighted mean and pooled standard deviation; and (3) because three outcome measures fell into our accuracy operationalization of performance, we aggregated them into a single accuracy indicator.

In three other studies, we estimated the effect size based on the result of the analysis of variance with one degree of freedom at between-group level [82, 83] or based on the p-value of a $\chi 2$ test and sample size [57].

Besides the weighted average effect size for each outcome, we computed the between-studies heterogeneity. Following the recommendations made by Borenstein et al. [84], we reported the Q, $I^2$, and $\tau^2$ indices. A statistically significant Q reflects if the true effect sizes vary across studies, whereas $\tau^2$ estimates the between-studies variance. Based on the standard deviation of true effects ($\tau$), a prediction interval for the true effects can be estimated, informing how widely the effects vary. Moreover, the $I^2$ statistic quantifies the dispersion observed due to true variations in effect sizes.

## Additional analyses

**Detecting extreme effect sizes.**   A single sample may heavily influence the results and conclusion of a meta-analysis if it is abnormally large, affecting the validity and robustness of the meta-analysis [85, 86]. We followed the approach proposed by Viechtbauer and Cheung [86] and defined a study as an outlier when its effect size estimate is so extreme that the study cannot be part of the "population" of effect sizes we pool in our meta-analysis, differing significantly from the overall effect. We examined outliers in each meta-analytic distribution as follows: (1) identifying outliers based on the study's confidence interval (i.e., if there are studies for which the confidence interval does not overlap with the confidence interval of the pooled effects), (2) we proceeded to analyze the data with and without the studies identified as being extreme, and (3) if one such extreme value had a substantial impact on the results (e.g., shifts the heterogeneity from non-significant to significant), we proceeded to remove it from the subsequent (main) analysis.

**Moderator analyses.**   The hypothesized categorial moderators were tested using subgroup analyses based on a mixed-effects model (i.e., employing a random-effects model within each subgroup, while between-subgroup differences were tested for significance based on a fixed-effect model). We used meta-regression under a random-effects model for numerical moderators.

**Publication bias.**    For this purpose, we corroborated information from multiple sources. After visually inspecting the forest plot to see the relationship between sample size and effects we also used Egger's test, which yields a *p*-value [87], as well as Duval and Tweedie's [88] trim and fill procedure to estimate the effect sizes after taking into account publication bias (i.e., imputed studies).

All meta-analytical analyses and publication bias were conducted with the aid of Comprehensive Meta-Analysis Version 3 software [89]. The prediction intervals were calculated with the spreadsheet provided by Borenstein et al. [84].

## Results

### Selection and inclusion of studies

The systematic search yielded 4868 unique records (after removing the duplicates) (see Fig 1). These were first reviewed based on title and abstract, removing another 4825 entries. In the next step, we analyzed the full text for 43 articles, two of which were included after screening the reference lists for potentially relevant studies. Out of these records, 25 were excluded for the following reasons: (1) 6 for not using a control group, (2) 9 employed a within-subjects design, and (3) 10 had incomplete data for computing effect sizes. The corresponding authors were contacted for this latter category, making available data from only one more study [81]. Hence, the final sample included 22 independent study samples that resulted from 19 publications (see Fig 1 for the complete flow of the literature search).

### Description of the included studies

A systematic overview of the studies' characteristics (e.g., participants, design) is displayed in Table 1. A summary of attributes of experimental manipulations (e.g., breaks, tasks before breaks) and outcomes are shown in Table 2. Moreover, a detailed narrative overview is provided in S1 File.

Eleven studies used samples of students, whereas the other ten studies tested their hypotheses on samples of employees. One study used "normal volunteers" to characterize its sample without further clarifications. The total number of participants in these studies was 2335, with a mean age of 31.2 years old.

Most of the studies employed an experimental design, where participants were randomly allocated to different interventions (*n* = 15), whereas fewer used non-random allocation but with equivalent groups (*n* = 7). Most of the studies took place in a laboratory (*n* = 13); a smaller number was in an organizational/workplace setting (*n* = 9). In most studies, participants in the control group were engaged in some break or free time between work tasks (*n* = 12). In contrast, in a slightly smaller number of studies, these participants continued working without respite (*n* = 10).

To study the resource replenishing effects of breaks, participants had to complete a series of tasks before taking a respite. These tasks were either relevant to organizational life, such as work simulations and actual work-related tasks (*n* = 13), or irrelevant, such as various cognitive tests (*n* = 9). Participants were exposed to different type of demands, such as cognitive (*n* = 7), emotional (n = 3), or clerical (*n* = 8). Studies in which participants had to generate new ideas were considered as having creative demands (*n* = 3). Time on task is one of the most widely studied contributors to the depletion of resources. In our sample, this information was specified only for 13 studies, with participants spending between a minimum of 2 minutes and a maximum of 4 hours on resource-demanding tasks before getting a break.

In almost all of the studies, the break was a non-work one (*n* = 19), whereas only one study included a work-related break. Two studies were incorporated into a category with work and

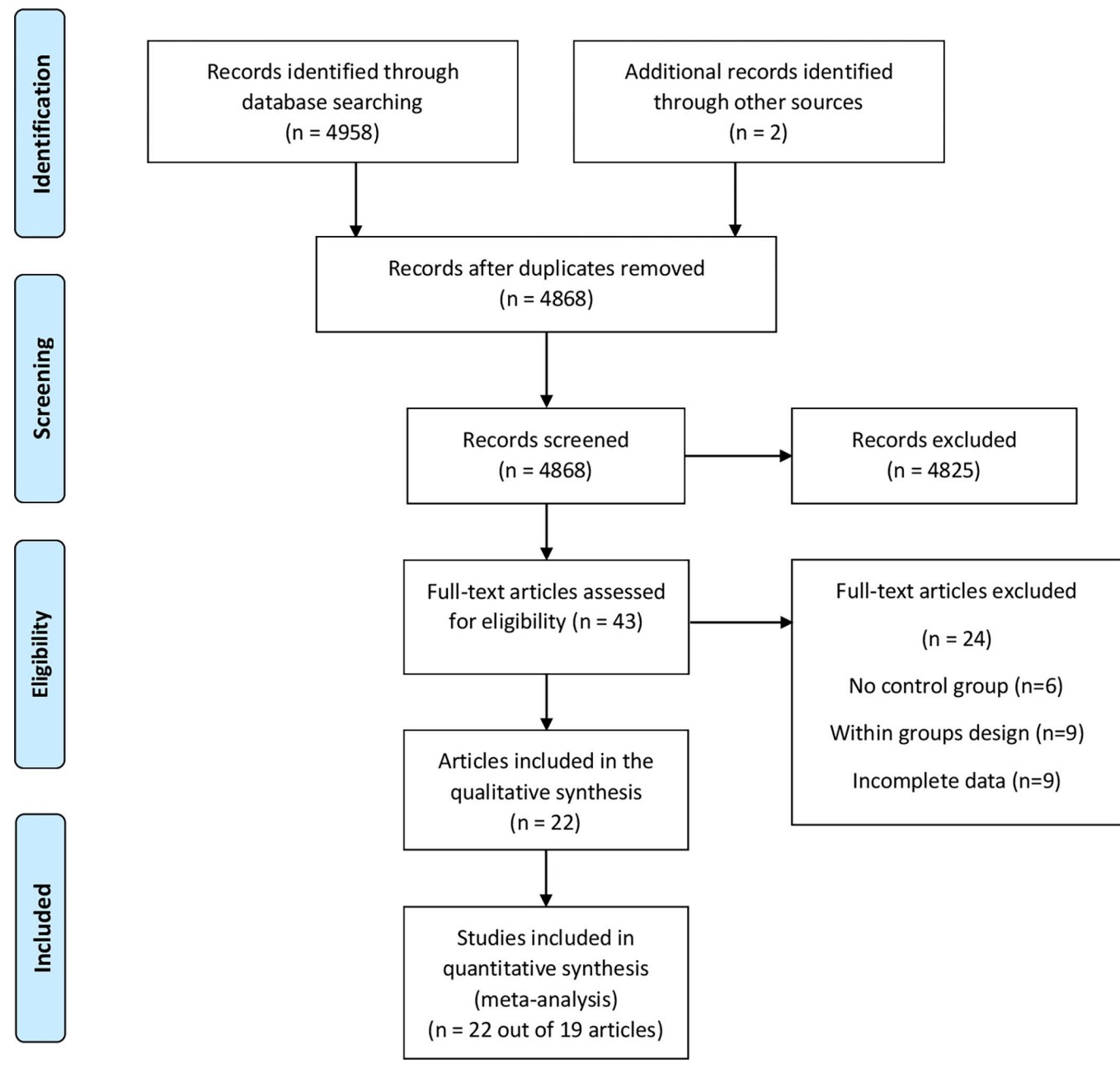

**Fig 1. The PRISMA flow diagram of the literature search and selection process.**

non-work micro-breaks because they shared the same control group, and their results were pooled in single indicators. Thus, the low variability on this factor also impedes treating it as a moderator (despite our initial intention). Regarding what participants did during the micro-breaks, five studies used a combination of two activities. For three of these studies, the combination of multiple types of activities was a result of using the same control group to compare different interventions. Five studies included a cognitive micro-break, where participants were involved in activities such as watching movie clips. Six studies offered a physical break between work bouts, whereas the other six activities during breaks were relaxing. Hence, the

**Table 1. Summary of studies included in the meta-analysis and their description.**

| Publication | Study no. | Country | Sample | Mean age | % Women | Randomization | Control | Setting | Risk of bias | | |
|---|---|---|---|---|---|---|---|---|---|---|---|
| | | | | | | | | | Low | Unclear | High |
| Bennett et al. (2020) [17] | | USA | Students | 24.2 | 55.2 | Yes | No break | Laboratory | 3 | 3 | 0 |
| Beute & de Kort (2014) [49] | Study 1 | Netherlands | Students | 22.2 | 53.3 | No | Break | Laboratory | 1 | 3 | 2 |
| Blake et al. (2019) [90] | | China | Employees | 32.5 | 47.5 | Yes | Break | Organization | 4 | 0 | 2 |
| Blasche et al. (2013) [75] | | Austria | Employees | 40.1 | 60.2 | Yes | Break | Organization | 2 | 2 | 2 |
| Clauss et al. (2018) [76] | | Germany | Employees | 42.3 | 71.1 | No | Break | Organization | 1 | 3 | 2 |
| Conlin et al. (2020) [77] | | USA | Students | 20 | 33.1 | No | No break | Laboratory | 2 | 4 | 0 |
| Ellwood et al. (2009) [78] | | Australia | Students | 22 | 72.2 | No | No break | Laboratory | 2 | 3 | 1 |
| Finstad et al. (2006) [91] | Study 1 | USA | Students | - | - | Yes | Break | Laboratory | 2 | 4 | 0 |
| | Study 2 | USA | Students | - | - | Yes | No break | Laboratory | 2 | 4 | 0 |
| | Study 3 | USA | Students | - | - | Yes | No break | Laboratory | 2 | 4 | 0 |
| Janicke et al. (2018) [79] | | USA | Employees | 36.2 | 47 | Yes | Break | Organization | 4 | 2 | 0 |
| Kennedy & Ball (2007) [82] | | Australia | Employees | 29.9 | 57.3 | No | Break | Organization | 2 | 1 | 3 |
| Lacaze et al. (2010) [57] | | Brazil | Employees | 30 | 73.4 | No | Break | Organization | 2 | 3 | 1 |
| Michishita et al. (2017) [50] | | Japan | Employees | 45 | 23.8 | Yes | Break | Organization | 1 | 3 | 2 |
| Michishita et al. (2017) [51] | | Japan | Employees | 40.9 | 32.2 | Yes | Break | Organization | 1 | 4 | 1 |
| Paulus et al. (2006) [80] | Study 2 | USA | Students | | | Yes | No break | Laboratory | 1 | 5 | 0 |
| | Study 3 | USA | Students | | | Yes | No break | Laboratory | 1 | 5 | 0 |
| Rees et al. (2017) [81] | | Australia | Students | 20.5 | 33.3 | Yes | No break | Laboratory | 2 | 4 | 0 |
| Rieger et al. (2017) [59] | | Germany | Students | 25.5 | 72.7 | Yes | Break | Laboratory | 3 | 3 | 0 |
| Steidle et al. (2017) [92] | | Germany | Employees | 36.9 | 43.9 | Yes | Break | Organization | 2 | 3 | 1 |
| Steinborn & Huestegge, (2016) [83] | | Germany | Unknown | 21.7 | 85 | No | No break | Laboratory | 1 | 4 | 1 |
| Wollseiffen et al. (2016) [93] | | Germany | Employees | 41 | 46 | Yes | No break | Laboratory | 1 | 5 | 0 |

heterogeneous and mixed nature of the micro-breaks employed in the included studies impedes us from exploring this aspect as a potential moderator. Break duration varied significantly between studies, ranging between 8 seconds and 10 minutes.

Considering the outcomes' operationalizations, vigor and fatigue were exclusively assessed with self-report scales. The measures varied from multi-factor well-established instruments (e.g., Activation-Deactivation-Checklist [94]; Utrecht Work Engagement Scale [95]; Profile of Mood States, [96]), to one-item ones [e.g., 92] The reporting of reliability estimates was inconsistent across studies (and not possible for single-item measures), but when mentioned, generally exceeded $\alpha = .80$ or even $\alpha = .90$ [e.g., 17, 79, 92]. As for performance, there were studies using both objective and subjective measures of performance, as well as measures of creativity. We grouped accuracy (e.g., mean correct responses, number of words remembered, errors reported during the completion of tasks, etc.) and reaction speed in cognitive tasks into a category of objective measures ($n = 8$), self-perceived performance was considered as a subjective measure ($n = 4$), and idea generation was considered as creative performance ($n = 3$). A more fine-grained categorization (especially for the objective measures) was hard to impose because

**Table 2. Characteristics of the interventions and outcomes.**

| Identification | Study | Intervention type | Characteristics of the task preceding the break | | | | | Characteristics of micro-breaks | | Outcome |
|---|---|---|---|---|---|---|---|---|---|---|
| | | | Type of task | Workplace relevance | Activity performed | Time on task (minutes) | | Break activity | Break duration (minutes) | |
| Bennett et al. (2020) [17] | | Work & Non work | Clerical | Relevant | Attention Network Test (Fan, McCandliss, Sommer, Raz, & Posner, 2002) | 10 | | Cognitive & relaxation | 5 | Fatigue |
| | | | | | | | | | | Vigor |
| Beute & de Kort (2014) [49] | Study 1 | Non work | Clerical | Relevant | Typing task | – | | Cognitive & nature | 3 | Performance |
| | | | | | | | | | | Fatigue |
| | | | | | | | | | | Vigor |
| Blake et al. (2019) [90] | | Non work | Clerical | Relevant | Office tasks | – | | Physical | 10 | Performance |
| Blasche et al. (2013) [75] | | Non work | Clerical | Relevant | Office tasks | – | | Physical | 8 | Fatigue |
| Clauss et al. (2018) [76] | | Work | Emotional | Relevant | Nursing tasks | 210 | | Cognitive | 10 | Fatigue |
| Conlin et al. (2020) [77] | | Non work | Clerical | Relevant | Clerical editing task | 10 | | Cognitive & nature | 0.66 | Performance |
| Ellwood et al. (2009) [78] | | Work & non work | Creative | Irrelevant | Idea generation test | 2 | | Cognitive | 5 | Performance |
| Finstad et al. (2006) [91] | Study 1 | Non work | Cognitive | Irrelevant | Prospective memory test (Thorndike & Lorge, 1944) | 30 | | Relaxation Cognitive | 0.133 | Performance |
| | Study 2 | Non work | Cognitive | Irrelevant | Prospective memory test (Thorndike & Lorge, 1944) | 30 | | Relaxation | 0.166 | Performance |
| | Study 3 | Non work | Cognitive | Irrelevant | Prospective memory test (Thorndike & Lorge, 1944) | 30 | | Relaxation | 0.166 | Performance |
| Janicke et al. (2018) [79] | | Non work | | Relevant | – | 240 | | Cognitive | 4 | Vigor |
| Kennedy & Ball, (2007) [82] | | Non work | Emotional | Relevant | Call center tasks | – | | Relaxation | 10 | Fatigue |
| | | | | | | | | | | Vigor |
| Lacaze et al. (2010) [57] | | Non work | Emotional | Relevant | Call center tasks | 180 | | Physical | 10 | Fatigue |
| | | | | | | | | | | Performance |
| Michishita et al. (2017) [50] | | Non work | Clerical | Relevant | – | – | | Physical | 10 | Fatigue |
| | | | | | | | | | | Vigor |
| Michishita et al. (2017) [51] | | Non work | Clerical | Relevant | – | – | | Physical | 10 | Fatigue |
| | | | | | | | | | | Vigor |
| Paulus et al. (2006) [80] | Study 2 | Non work | Creative | Irrelevant | Brainstorming | 16 | | Relaxation | 4.5 | Performance |
| | Study 3 | Non work | Creative | Irrelevant | Brainstorming | 16 | | Relaxation | 4.5 | Performance |
| Rees et al. (2017) [81] | | Non work | Cognitive | Relevant | Simulated rail control task | 20 | | Cognitive | 5 | Performance |
| Rieger et al. (2017) [59] | | Non work | Cognitive | Irrelevant | Reading-span task (Daneman & Carpenter, 1980) and Operation-span task (Turner & Engle, 1989) | – | | Cognitive | 2 | Vigor |
| | | | | | | | | | | Performance |
| Steidle et al. (2017) [92] | | Non work | Clerical | Relevant | Office tasks | – | | Nature & physical | 10 | Fatigue |
| | | | | | | | | | | Vigor |
| Steinborn & Huestegge, (2016) [83] | | Non work | Cognitive | Irrelevant | Mental-addition and verification tasks (Zbrodoff & Logan, 1990) | – | | Cognitive & physical | 3 | Performance |

*(Continued)*

**Table 2.** (Continued)

| Identification | Study | Intervention type | Characteristics of the task preceding the break | | | | Characteristics of micro-breaks | | Outcome |
|---|---|---|---|---|---|---|---|---|---|
| | | | Type of task | Workplace relevance | Activity performed | Time on task (minutes) | Break activity | Break duration (minutes) | |
| Wollseiffen et al. (2016) [93] | | Non work | Cognitive | Irrelevant | Memory matrix (Dorval & Pepin, 1986; Schaefer & Thomas, 1998), and Chalkboard challenge d2-R test (R Brickenkamp, 2002; Rolf Brickenkamp, Schmidt-atzert, & Liepmann, 2010) | 120 | Physical | 3 | Fatigue |
| | | | | | | | | | Vigor |

of the limited number of studies. In terms of reliability, with two exceptions (Studies 2 and 3 from Paulus et al. [80]), there were no such estimates reported. While the mentioned cases focused on creative performance with interrater reliabilities of > .90, for all the other operationalizations it was not possible to make any accurate inference from this perspective. However, as also previously discussed [70], the objective measures typically imply the employment of arbitrary tasks and/or unaudited performance measures or paradigms with reduced numbers of trials. Hence, it is less likely to expect such approaches as being psychometrically precise. Unfortunately, in the absence of reliability estimates, such measurement artefacts cannot be taken into account meta-analytically.

## Quality of the included studies

The risk of bias assessment results for each threat to the internal validity are shown in Table 1 (for each study) and Fig 2 (as a visual summary).

The results following the selection bias assessment show an unclear risk for both domains (i.e., sequence generation and allocation concealment), as thirteen of the twenty-two studies did not make explicit the randomization method. Detection bias follows the trend, with most studies in the unclear risk of bias category (72% of included studies). Most studies were at low risk for attrition bias ($n = 20$). Twenty-one studies were assessed with unclear risk regarding reporting bias. For other potential threats to validity, fifteen studies out of twenty-two have been evaluated as having low risk.

Overall, only four out of twenty-two studies were assessed with a low risk of bias, whereas one presented a high risk for at least half of the criteria. Thus, we are inclined to consider the risk of bias in our overall sample as being somewhat unclear.

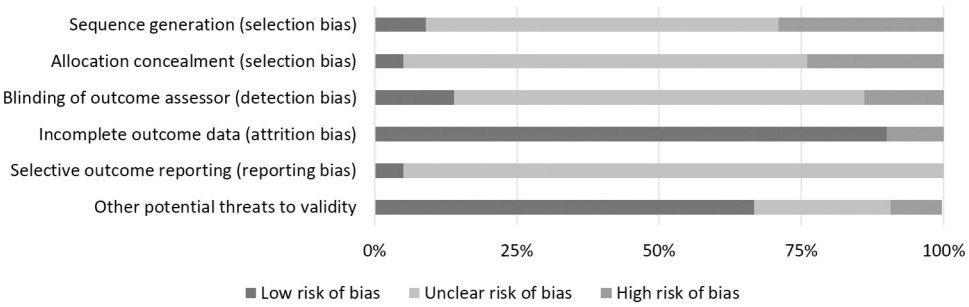

**Fig 2. Risk of internal bias summary.**

## Preliminary analyses

To detect potential outliers, we analyzed the studies for which the confidence intervals failed to overlap with the confidence interval of the pooled effects. We found only two potential outliers. One of them was in the sample of effects regarding the role of micro-breaks on fatigue [49], with an effect size of $d$ = 13.43, and a 95%CI which stretched between 11.11, and 15.75, being completely non-overlapped with the overall one [0.29, 1.52]. The other one was in the sample of effects of micro-breaks on performance outcomes [77], with an effect size of $d$ = 3.85, 95%CI of [3.38, 4.33], as compared to the meta-analytical one laying between [-0.14, 0.86]. The decision to exclude them was proven legitimate, as the results differed greatly, with improvements in heterogeneity. Specifically, for the effect on fatigue; before exclusion the heterogeneity was statistically significant ($Q_{(9)}$ = 127.55, $p$ < .001), while after excluding the outlier became non-significant ($Q_{(8)}$ = 6.24, $p$ = .619), result also reflected into the $I^2$ value (before exclusion: $I^2$ = 92.94%; and after: $I^2$ = 0.00%). In the case of performance, heterogeneity with the outlier was also higher ($Q_{(15)}$ = 242.57, $p$ < .001; $I^2$ = 93.82%), as without it ($Q_{(14)}$ = 34.52, $p$ = .002; $I^2$ = 59.45%). The detailed analyses including the outliers can be found in S2 Table.

## Effectiveness of micro-breaks

The main meta-analytical results are presented in Table 3 and displayed in Fig 3. These revealed a statistically significant but small effect of micro-breaks on vigor, $d$ = 0.36, $p$ < .001, 95% CI [.16, .55], and fatigue $d$ = 0.35, $p$ < .001, 95% CI [.19, .50], while the effect on performance was not statistically significant, $d$ = 0.16, $p$ = .17, 95% CI [-0.04, .37].

Heterogeneity analyses suggested that the effects on vigor ($Q_{(8)}$ = 13.61, $p$ = .093; $I^2$ = 41.21%; $\tau^2$ = .04; 95% prediction interval: [-.15, .86]) and especially on fatigue ($Q_{(8)}$ = 6.25, $p$ = .619; $I^2$ = 0.00%; $\tau^2$ = .00; 95% prediction interval: [.16, .53]) were quite homogenous. However, there was significant unexplained variance in the true effect sizes for performance ($Q_{(14)}$ = 34.52, $p$ = .002; 95% prediction interval: [-.53, .86]). A big proportion of the observed variance being due to real variations of the effects ($I^2$ = 59.45%, $\tau^2$ = 0.09).

## Moderator analyses

Even though the effects on vigor and fatigue were relatively homogeneous, and only those on performance revealed significant between-studies variations, we continued with the moderator analyses on all three outcomes (especially since we still had rational/theoretical arguments in this regard). The results of the moderator analyses for all outcomes of interest can be seen in Table 4. Because of the modest number of studies on each category and the low variability in many cases, we only had the methodological possibility to test a low number of moderators

**Table 3. Effectiveness of micro-breaks on vigor, fatigue, and performance.**

| Outcome | k | n1 | n2 | d | SE | 95% CI | p | Q | $\tau^2$ | $I^2$ | 95% prediction interval |
|---|---|---|---|---|---|---|---|---|---|---|---|
| Vigor | 9 | 614 | 299 | .36 | 0.10 | [.16, .55] | < .001 | 13.61 | 0.04 | 41.21 | [-.15, .86] |
| Fatigue | 9 | 528 | 275 | .35 | 0.08 | [.19, .50] | < .001 | 6.25 | 0.00 | 0.00 | [.16, .53] |
| Performance | 15 | 711 | 421 | .16 | 0.10 | [-.04, .37] | .116 | 34.52** | 0.09 | 59.45 | [-.53, .86] |

$k$ = number of studies included in the analysis; n1 = number of participants included in the intervention groups, n2 = number of participants included in the control groups; $d$ = weighted average effect size; $SE$ = standard error of the average effect size; 95% CI = 95% confidence interval; $Q$ = statistical test for the estimation of heterogeneity; $\tau^2$ = between-study variance; $I^2$ = proportion of variation in the observed that is due to true effects variation (%).

*p < .05

**p < .01

## (a) Vigor

| Study name | Statistics for each study | | | | | | | Std diff in means and 95% CI |
|---|---|---|---|---|---|---|---|---|
| | Std diff in means | Standard error | Variance | Lower limit | Upper limit | Z-Value | p-Value | |
| Bennett et al. 2020 | 0.288 | 0.212 | 0.045 | -0.128 | 0.704 | 1.358 | 0.175 | |
| Beute & de Kort 2014 (Study 1) | 1.017 | 0.293 | 0.086 | 0.442 | 1.592 | 3.468 | 0.001 | |
| Janicke et al. 2018 | 0.305 | 0.169 | 0.029 | -0.026 | 0.637 | 1.804 | 0.071 | |
| Kennedy & Ball, 2007 | 0.160 | 0.234 | 0.055 | -0.298 | 0.618 | 0.684 | 0.494 | |
| Michishita et al. 2017 (a) | 0.692 | 0.181 | 0.033 | 0.338 | 1.046 | 3.832 | 0.000 | |
| Michishita et al. 2017 (b) | 0.057 | 0.260 | 0.068 | -0.454 | 0.567 | 0.218 | 0.828 | |
| Rieger et al. 2017 | 0.000 | 0.213 | 0.045 | -0.417 | 0.418 | 0.002 | 0.998 | |
| Steidle et al. 2017 | 0.417 | 0.232 | 0.054 | -0.037 | 0.871 | 1.799 | 0.072 | |
| Wollseiffen et al. 2016 | 0.365 | 0.451 | 0.203 | -0.518 | 1.249 | 0.810 | 0.418 | |

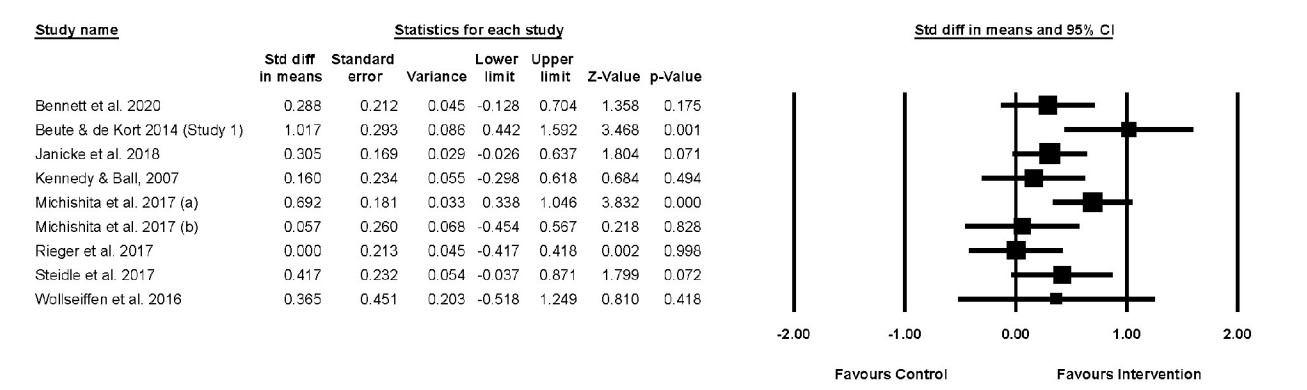

## (b) Fatigue

| Study name | Statistics for each study | | | | | | | Std diff in means and 95% CI |
|---|---|---|---|---|---|---|---|---|
| | Std diff in means | Standard error | Variance | Lower limit | Upper limit | Z-Value | p-Value | |
| Bennett et al. 2020 | 0.227 | 0.212 | 0.045 | -0.188 | 0.643 | 1.072 | 0.284 | |
| Blasche et al. 2013 | 0.319 | 0.261 | 0.068 | -0.192 | 0.830 | 1.224 | 0.221 | |
| Clauss et al. 2018 | 0.275 | 0.239 | 0.057 | -0.193 | 0.743 | 1.151 | 0.250 | |
| Kennedy & Ball, 2007 | 0.240 | 0.234 | 0.055 | -0.219 | 0.699 | 1.024 | 0.306 | |
| Lacaze et al. 2010 | 0.900 | 0.262 | 0.069 | 0.386 | 1.414 | 3.431 | 0.001 | |
| Michishita et al. 2017 (a) | 0.454 | 0.178 | 0.032 | 0.106 | 0.802 | 2.555 | 0.011 | |
| Michishita et al. 2017 (b) | 0.150 | 0.261 | 0.068 | -0.361 | 0.661 | 0.575 | 0.565 | |
| Steidle et al. 2017 | 0.274 | 0.230 | 0.053 | -0.177 | 0.725 | 1.190 | 0.234 | |
| Wollseiffen et al. 2016 | 0.178 | 0.448 | 0.201 | -0.701 | 1.056 | 0.397 | 0.692 | |

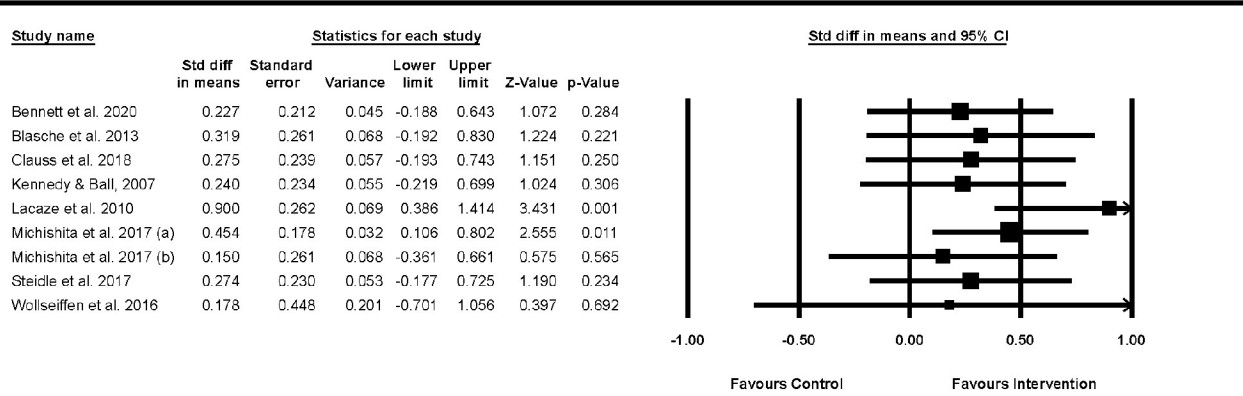

## (c) Performance

| Study name | Statistics for each study | | | | | | | Std diff in means and 95% CI |
|---|---|---|---|---|---|---|---|---|
| | Std diff in means | Standard error | Variance | Lower limit | Upper limit | Z-Value | p-Value | |
| Beute & de Kort 2014 (Study 1) | 0.923 | 0.308 | 0.095 | 0.319 | 1.527 | 2.997 | 0.003 | |
| Blake et al. 2019 | 0.339 | 0.130 | 0.017 | 0.084 | 0.594 | 2.604 | 0.009 | |
| Ellwood et al. 2009 | 0.299 | 0.225 | 0.050 | -0.141 | 0.739 | 1.331 | 0.183 | |
| Finstad et al. 2006 (Study 1) | -0.086 | 0.321 | 0.103 | -0.714 | 0.542 | -0.269 | 0.788 | |
| Finstad et al. 2006 (Study 1) | -0.285 | 0.304 | 0.093 | -0.881 | 0.312 | -0.935 | 0.350 | |
| Finstad et al. 2006 (Study 2) | -0.645 | 0.363 | 0.132 | -1.356 | 0.065 | -1.780 | 0.075 | |
| Finstad et al. 2006 (Study 2) | -0.470 | 0.358 | 0.128 | -1.173 | 0.232 | -1.313 | 0.189 | |
| Finstad et al. 2006 (Study 3) | -0.307 | 0.300 | 0.090 | -0.895 | 0.281 | -1.024 | 0.306 | |
| Finstad et al. 2006 (Study 3) | -0.552 | 0.315 | 0.099 | -1.169 | 0.065 | -1.755 | 0.079 | |
| Lacaze et al. 2010 | 0.565 | 0.255 | 0.065 | 0.065 | 1.065 | 2.216 | 0.027 | |
| Paulus et al. 2006 (Study 2) | 0.514 | 0.242 | 0.059 | 0.038 | 0.989 | 2.119 | 0.034 | |
| Paulus et al. 2006 (Study 3) | 0.330 | 0.243 | 0.059 | -0.146 | 0.807 | 1.358 | 0.174 | |
| Rieger et al. 2017 | 0.559 | 0.217 | 0.047 | 0.134 | 0.984 | 2.578 | 0.010 | |
| Steinborn & Huestegge, 2016 | 0.185 | 0.245 | 0.060 | -0.296 | 0.665 | 0.754 | 0.451 | |
| Rees et al. 2017 | 0.359 | 0.286 | 0.082 | -0.201 | 0.919 | 1.256 | 0.209 | |

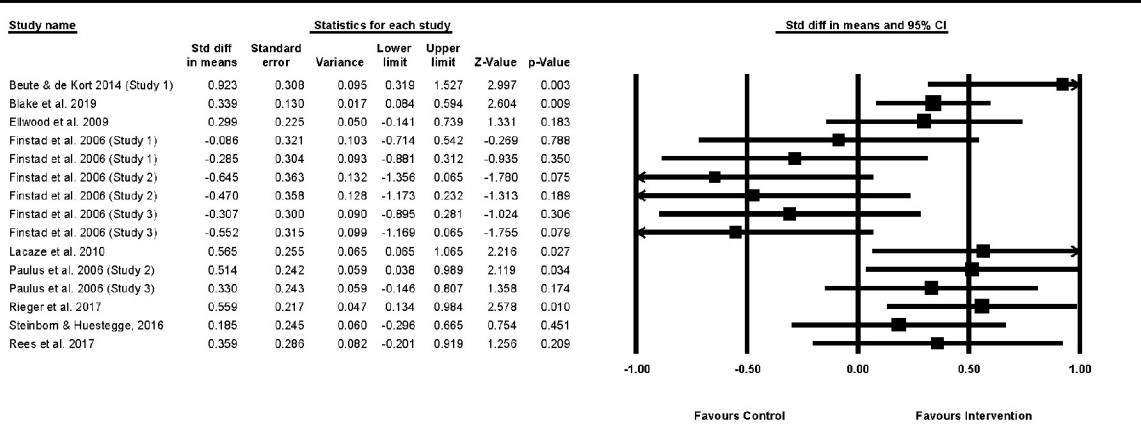

**Fig 3.** Standardized effect sizes and forest plot for the sample of studies regarding (a) Vigor, (b) Fatigue, and (c) Performance.

**Table 4. Test of significance for each presumed moderator.**

| Moderator | Vigor | | | Fatigue | | | Performance | | |
|---|---|---|---|---|---|---|---|---|---|
| | k | Q | p | k | Q | p | k | Q | p |
| Break duration (*minutes*) | 9 | 0.01 | .909 | 9 | 0.61 | .436 | 15 | 7.50 | **.006** |
| Antecedent task (*cognitive vs. clerical vs. emotional vs. creative*) | 8 | 2.28 | .094 | 8 | 0.43 | .514 | 14 | 6.53 | **.011** |
| Professional category (*employees vs. students*) | 9 | 0.02 | .899 | 9 | 0.37 | .542 | 14 | 2.66 | .103 |
| Study setting (*laboratory vs. workplace*) | 9 | 0.02 | .902 | 9 | 0.55 | .459 | 15 | 2.73 | .099 |
| Type of control (*break vs. no break*) | 9 | 0.08 | .777 | 9 | 0.55 | .459 | 15 | 2.66 | .103 |
| Performance operationalization (*objective, subjective, creative*) | – | – | – | – | – | – | 15 | 4.31 | .116 |

The effect of break duration was tested with meta-regression while the other moderators were tested with subgroup analysis.

(i.e., break duration, the task before break, type of participants, study setting, or control group activity).

For vigor and fatigue, the results show that none of the considered moderators impacted the efficacy of micro-breaks (all $ps > .050$). For overall performance, however, two moderators were found. Break duration was revealed to be one of the significant results ($b = .07$, $p = .006$, $R^2 = .34$). Indicating that the longer the break, the more micro-break leads to a performance increase. The second significant effect was the type of task performed before the break ($Q_{(2)} = 6.53$, $p = .011$). More specifically, when the task was cognitive, the micro-breaks had a very small effect on performance ($d = -.09$, 95%CI [-.39, .30], $p = .541$), which was non-significant and still heterogeneous. When the tasks were creative, micro-breaks had a small, significant effect ($d = .38$, 95%CI [.11, .64], $p = .006$), and for clerical tasks the effect of micro-breaks was medium and significant ($d = .56$, 95%CI [0.01, 1.12], $p = .047$). It is important to mention that the latter effect is based only on two studies; hence, it has to be cautiously interpreted. To have a broader picture of the role of task antecedent to the break, in Table 5, we also report the subgroup results for vigor and fatigue (not only for performance). It may be worth noticing that the effect on vigor when the micro-break is taken from a cognitive task is also minimal and non-significant (important to bear in mind that the sub-groups differences are not statistically significant).

Finally, we tested if study quality (i.e., total criteria with low risk of bias for internal validity–see Table 1) was associated with the efficacy of the interventions. The results of the meta-

**Table 5. Meta-analytical findings at each level of the antecedent task moderator.**

| Outcome | Moderator levels | k | d | SE | 95%CI | p | Q | $\tau^2$ | $I^2$ |
|---|---|---|---|---|---|---|---|---|---|
| *Antecedent task (activity preceding the break)* | | | | | | | | | |
| Vigor | Cognitive | 2 | .07 | .19 | [-.31, .45] | .728 | 9.14 | .04 | 45.29% |
| | Clerical | 6 | .45 | .12 | [.21, .68] | < .001 | 0.54 | .00 | 0.00% |
| Fatigue | Clerical | 5 | .31 | .10 | [.11, .50] | .002 | 1.21 | .00 | 0.00% |
| | Emotional | 3 | .46 | .21 | [.05, .86] | .027 | 4.28 | .07 | 53.26% |
| Performance | Cognitive | 9 | -.09 | .15 | [-.39, .20] | .541 | 18.71* | .11 | 57.25% |
| | Clerical | 2 | .56 | .28 | [.01, 1.12] | .047 | 3.05 | .12 | 67.23% |
| | Creative | 3 | .38 | .14 | [.11, .64] | .006 | 0.47 | .00 | 0.00% |

*Note*: The analysis on vigor was done without the effect from Kennedy and Ball [81] being a single study with emotional labor task; the analysis on fatigue was performed without the effect from Wollseiffen et al. [93], being the only study with cognitive task; the analysis on performance was conducted without Lacaze et al. [57], because we could not accurately classify the type of task.

\* $p < .05$

regression suggest that for neither of the outcomes the amount of criteria with low risk of bias had a significant impact on efficacy (effect on vigor: $b = -.05$, $p = .642$, $R^2 = .00$; effect on fatigue: $b = -.01$, $p = .905$, $R^2 = .00$; effect on performance: $b = -.13$, $p = .329$, $R^2 = .00$).

## Publication bias

For the effect size estimates on vigor, Egger's test was not statistically significant (intercept = 0.21, $p = .915$), while the trim and fill procedure imputed two studies to the right of the mean. The adjusted effect was $d = 0.45$, being similar in magnitude to the observed value ($d = 0.35$). In the case of fatigue, the statistically non-significant results on Egger's test (intercept = -0.46, $p = .769$) indicated a lack of publication bias, but the trim and fill procedure imputed two studies to the right of the mean. The adjusted effect was $d = 0.40$, being also like the observed one $d = 0.34$. Finally, for performance, Egger's test was statistically significant (intercept = -2.63, $p = .048$), indicating possible publication bias. The trim and fill procedure imputed two studies to the right of the mean, where the adjusted value suggests a significant small effect ($d = 0.22$, 95% CI [0.02, 0.43]) as compared to the observed one ($d = 0.14$, 95% CI [-0.07, 0.36]). Even though statistically significant, it still is similar in magnitude. Overall, whereas some evidence for the presence of publication bias exists, it does not seem to be an impactful threat to the observed effects.

## Discussion

The main objective of the present meta-analysis was to examine the efficacy of micro-breaks (less than 10 min pauses from tasks) on individual outcomes such as well-being (i.e., increased vigor and decreased fatigue) and performance. Moreover, we also considered the influence of work demands and several contextual factors, such as professional category or study setting, on the role of micro-breaks for focus outcomes.

Our results revealed that micro-breaks are efficient in preserving high levels of vigor and alleviating fatigue. It seems that the effects are univocal and generalizable for the well-being outcomes. These were relatively homogeneous, and none of the included moderators were significant. Hence, the data suggest that micro-breaks may be a *panacea* for fostering well-being during worktime.

When it comes to performance, the data revealed some nuances. The break duration was a significant covariate of the effect of micro-breaks: the longer the break, the better the performance. Moreover, the type of task from which participants were taking the break also emerged as a significant moderator. Micro-breaks could significantly increase performance for clerical work or creative exercises and not for a cognitively demanding task. These results have both theoretical and practical implications.

Firstly, our results support a central assumption of the recovery literature, which states that engaging in recovery activities (what an individual does) leads to a recovered system (more energy, less fatigue, and better performance on some tasks). When no further demands are put on the individual, recovery is possible through a short break from the work tasks [8].

Secondly, break duration was essential in recovery and micro-breaks literature [8, 17]. A break is taken in order to replenish energy to achieve goals and performance. The link between goals and performance is ensured by attention [97], a key concept in cognitive psychology studies. The difference between micro-breaks, short breaks, and long breaks, can be related from this perspective to the functioning of the three attention networks: alerting, guidance, and executive control [98]. Studies show that if the first two fluctuate on a momentary basis, executive control benefits from greater stability [99] and it allows individuals to monitor their attention focus having an impact on behavioral self-regulation [100]. However, at the same

time, break duration is largely missing from previous recovery research, considering time either as a boundary condition or not discussing time at all [54]. This lack of focus on break duration resulted in high variability between studies regarding the length of such breaks. Moreover, as a result, practical answers tested empirically to attest whether micro-break duration matters for well-being or performance are missing. Thus, even if micro-breaks are understood as short breaks under 10 minutes in duration [27], the specific time of these micro-breaks was not yet established [17, 27]. When considering only performance as outcome, other scholars also noticed the difficulty of giving a universal answer to the question of how long a break should be in order to be effective [70]. The present study contributes to this body of literature, supporting the assumption that short breaks of close to ten minutes efficiently alleviate fatigue, increase energy, and boost (subjective / perceived) performance. These results offer, therefore, some clarity related to a duration standard.

Thirdly, regarding the effects of micro-breaks on performance as a function of the task participants were engaged in, the results show that especially for clerical (routine tasks) or creative (where divergent thinking is needed) tasks, taking short breaks helps individuals in performing better at subsequent tasks. These results are in line with another meta-analysis showing that when attempting creative problems requiring a wider search of knowledge, individuals benefit from a period of time in which the problem is set aside prior to further attempts to solve it [101]. The effect of breaks also tends to be better reflected in subjective evaluations of performance and actual creative outputs. Therefore, micro-breaks make individuals feel more vigorous and less fatigued and stimulate them to feel more productive after the break.

Routine tasks refer to sequences of actions that are performed with a high level of automaticity, with high speed and low variability [102, 103]. They release cognitive resources to think about other aspects of the work or simply for the mind to wander, increasing the probability of making mistakes [104]. The break can decrease this risk, interrupt spontaneous ideas that cross to mind rapidly and unconsciously [105] and refocus the attention on the next task, facilitating performance.

Referring to creative tasks, they essentially demand creative cognition and divergent thinking [106]. Creative cognition supposes the ability to strategically search memory for task-relevant information and to suppress interference with other information that comes to mind during divergent thinking [107]. It is also connected with associative processes [108]. According to the dual-process model of creativity, there are two pathways to creative performance: flexibility and persistence. The flexibility pathway stimulates creativity by flexible switching between approaches and sets [109]. Thus, task switching strengthens flexibility and can further improve creative performance. If we look at the break as a switching task, this is a possible explanation for improving creative performance after the break by strengthening flexibility.

However, when it comes to cognitively demanding tasks, taking short breaks does not seem to affect subsequent performance. Based on recent experimental research showing that the break duration is an essential factor in understanding the recovery processes [17], a possible explanation for the result of our study is that the pause with a duration of less than 10 minutes can replenish vigor, but not fully restore the resources needed to perform in a demanding cognitive task. At the same time, our data also showed that when categorizing performance in subtypes, only the effects for cognitive performance still remain heterogeneous (variation for which we did not have sufficient data to further disentangle). As also Schumann et al. [70] pointed out in their review, there are multiple methodological factors (e.g., employment of arbitrary tasks and/or unreliable or unaudited performance measures; insufficient number of trials) which could result in artefactual variance. Moreover, even if taking into account their purpose, all these measures can be grouped under the umbrella term of performance, the many conceptual and operational differences between them could also make one rightfully

argue against their aggregation. Hence, these particular results should be taken with increased caution. As cognitive demands are prevalent in the workplace and also the educational settings, future studies might test the conditions under which respites positively affect subsequent performance by applying more standardized experimental paradigms and increasing consensus on methodological approaches.

Indeed, the effect of recovery during micro-breaks on subsequent performance seems to be more entangled compared to the effect it has on well-being, thus deserving a closer look. As a detailed inspection of aspects such as break length and the type of task from which the break is taken offered valuable insights but no definitive answers regarding this effect, having an overview also of the excluded studies could prove fruitful. Thus, we had another look at all the articles assessed for eligibility in the present meta-analysis, and specifically at those focused on performance outcomes. Whereas some laboratory studies found no effect of taking short breaks between various resource-depleting tasks on subsequent performance [e.g., 110–112], others found significant improvements in performance after taking a break [e.g., 57, 78, 83].

Studies in which various work tasks or environments were simulated also found either no significant impact of breaks on performance [113–115], or either positive effects [81]. For example, breaks were found to improve performance in a simulated rail control task in comparison to the control condition [81], whereas other studies simulating night shifts [113, 114] or simulating nighttime flights for piloting crew [115] found no such positive, significant effects. When considering experimental studies conducted only in real-life work settings, the patterns of results are similar. For example, a study introduced brief Qigong exercise breaks as interventions in two sites of a major organization in China for twelve consecutive weeks, finding no improvement in self-reported work performance for the office workers [90]. However, another study found positive effects of breaks on worker productivity for a similar population, namely computer operators of a large insurance company [56]. Moreover, two other studies found that intra-operative micro-breaks during which surgeons executed mild stretching exercises positively impacted their performance without affecting the operative duration [116, 117]. Importantly, no study found significantly decreased performance when a break was introduced between tasks, adding to the argument that even when less time is spent on the task at hand due to the time allocated for the break [19, 26], performance does not worsen compared to individuals who continued working [90, 112, 118]. In other words, even if we dismiss all the studies finding a positive impact of breaks on task efficiency and consider micro-breaks as not improving performance, taking a break at least does not harm it. Also, the boost in performance may be occupation or task specific (e.g., computer operators [56], or even surgeons [116, 117]), an avenue of research that further deserves scholarly focus.

From a practical perspective, these results offer strong support that taking short breaks during working hours is beneficial for individuals' health and productivity. Sedentary activities requiring constant monitoring and attention resulting from accelerated automation and the COVID-19 pandemic (e.g., online education, shift to remote working) might remain problematic. Taking short breaks can become more necessary to protect individual well-being and performance [119]. Therefore, organizations must reconsider the usefulness of an "always-on" culture for personal and organizational outcomes.

Managers can support employees' well-being by encouraging them to take micro-breaks. Such leadership engagement is relevant, considering that many employees still might feel that taking breaks might be perceived as counterproductive behavior [120]. Moreover, organizations could also benefit from training to build personal resources and organizational capacities, learning how and when to engage in efficient energy management and recovery strategies. We agree with the proposition of Bennett et al. [28] that further exploration of organizational

policies conducive for employee recovery would be very interesting for research and practice alike.

As mentioned before, about half of the studies were conducted on students. This aspect has several implications. Students are considered similar to employees in terms of their engagement in structured and constraining activities to a certain measure and directed towards specific goals, such as completing assignments and attending classes [121, 122]. Also, their experience with various demands and associated distress emphasizes the need to understand how detrimental effects, such as fatigue, can be prevented, and positive outcomes, such as vitality and performance, can be enhanced. Therefore, how micro-breaks are structured and experienced can significantly influence students' well-being and performance outcomes.

Moreover, our results can also have implications for an educational perspective. Lectures can become more successful when delivering information alternate with learning pauses [123, 124]. In this context, restorative breaks, especially in online learning, where students need to look away from their computers may help reestablish their energy and focus. In this way, it is possible to have a deep learning process right in the educational setting and an enhanced chance to sustain performance in the longer term. Teaching students the benefits of short breaks during individual study for optimal learning can be one of the goals of educational policies that increase students' motivation and achievements. However, one should bear in mind that these observations are only presumed implications for the academic environment, being based on activities more specific to the workplace environment. As previously mentioned [125], even though academic classwork, as also office work, is a source of fatigue, the two environments bear differences that make the generalization between them cautionary.

## Limitations, future research directions, and conclusions

Although this meta-analysis sheds some light on the effect of micro-breaks on individual outcomes, it is not without any limitations, thus prompting the reader to interpret these results with caution.

Firstly, the number of studies was modest for each outcome, which limited the moderator analyses. Specifically, essential moderators that could have helped us better understand the heterogeneity presented for the performance effect were impossible to analyze. Some examples are break type (work or non-work), specific activity performed during the break (physical, relational, cognitive, relaxation, nature-related, etc.), or time working on the task before taking a break. In the recent literature review on the effect of breaks on performance, the aspect of testing moderator variables was also discussed, concluding that underdeveloped methodological approaches for tasks and measures, and extremely heterogeneous perspectives concerning contexts, and designs, make it hard to conduct such differential analyses in the field of rest-break research [70]. Thus, we could not address one of the most important questions for practice about which specific activity is most efficient for recovering lost resources during work: "What to do in these breaks to feel and perform better?". However, while we still need a clear explanation for the performance outcome, at least for well-being, the answer seems to be "any type of decoupling activity".

Secondly, besides the increased heterogeneity in measurement approaches (especially when considering performance), another important cautionary aspect may be their reliability. The fact that we have inconsistent reporting for the well-being self-reports and no information for the other measures makes it impossible to scrutinize the potential bias induced by measurement error. When possible, future studies should not overlook reporting reliability estimates. This way updated meta-analyses on this topic may address the artefactual variance accountable by measurement error [126, 127].

Furthermore, because all the studies considered in the present meta-analysis used self-reports of well-being outcomes, our results could be exposed to response biases [128]. Although this is a well-known risk when using self-reports, the energetic activation (e.g., vigor) and deactivation (e.g., fatigue) of the well-being components considered in this meta-analysis represent a subjective component of a "bio-behavioral system of activation" [129, 130 p. 827], experienced as feelings, emotions, or dispositions [7]. Thus, one's evaluation of their subjective experience can be more suitable than other individuals' reports or assessments. However, future studies could also benefit from a combination of self-reports of well-being with objective measures of the bio-behavioral systems involved (e.g., endocrinological indicators) [130].

Lastly, we considered only two aspects of well-being, a pleasant activation (e.g., vigor) and unpleasant deactivation (e.g., fatigue), leaving aside other concepts studied in recovery research such as anxiety or tension [131]. Moreover, the present meta-analysis is based on studies from the pre-pandemic literature. The new Covid-19 pandemic brought forward new challenges at home, at school, and at work, changing the way people work in fundamental ways [132]. Thus, future studies could benefit from including this new challenge related to the pandemic in their designs.

The current meta-analysis was performed on twenty-two experimental studies published in the past thirty years that tested the effects of micro-breaks on vigor, fatigue, and performance. Specifically, it showed that micro-breaks positively impact well-being by enhancing vigor and lowering fatigue, regardless of the contextual factors. Importantly, micro-breaks do not seem to influence performance generally. However, when the break is more extended, the performance tends to improve, especially when individuals are engaged in creative or clerical tasks, and less when performing activities of a cognitively demanding nature.

## Supporting information

**S1 Checklist. PRISMA 2020 checklist.**
(DOCX)

**S1 Table. Details regarding the search string for each database.**
(DOCX)

**S2 Table. Overall effect on fatigue and performance with the outliers included.**
(DOCX)

**S1 File. Detailed description of the included studies.**
(DOCX)

## Author Contributions

**Conceptualization:** Patricia Albulescu, Coralia Sulea.

**Data curation:** Andrei Rusu, Alexandra Bodnaru.

**Formal analysis:** Andrei Rusu, Alexandra Bodnaru.

**Methodology:** Patricia Albulescu, Andrei Rusu, Alexandra Bodnaru.

**Project administration:** Bogdan Tudor Tulbure.

**Resources:** Irina Macsinga, Andrei Rusu, Coralia Sulea, Bogdan Tudor Tulbure.

**Software:** Andrei Rusu.

**Supervision:** Andrei Rusu, Coralia Sulea, Bogdan Tudor Tulbure.

**Visualization:** Irina Macsinga, Coralia Sulea, Alexandra Bodnaru, Bogdan Tudor Tulbure.

**Writing – original draft:** Patricia Albulescu.

**Writing – review & editing:** Patricia Albulescu, Irina Macsinga, Andrei Rusu, Coralia Sulea, Alexandra Bodnaru, Bogdan Tudor Tulbure.

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
