## [Decision Letter · Decision Letter 0]

2 May 2022

PONE-D-22-07400Let's take a break! A systematic review and meta-analysis on the efficacy of micro-breaks for increasing well-being and performancePLOS ONE

Dear Dr. Macsinga,

Thank you for submitting your manuscript to PLOS ONE. After careful consideration, we feel that it has merit but does not fully meet PLOS ONE’s publication criteria as it currently stands. Therefore, we invite you to submit a revised version of the manuscript that addresses the points raised during the review process. Editorial comment: Two expert reviewers commented on your manuscript. As you can see, both referees seem to consider you work important overall, while at the same time provided a whole number of additional points that might be useful to consider during the revision of the manuscript. To name the most important points, both referees raised the methodical points of how to best quantify study results, and when to consider a qualitative evaluation of aspects of the studies. Here, both refer to findings on the connection between experience and performance (which sometimes go together, othertimes not), and, to the reliability of the underlying tests and performance measures. My personal suggestion is to also consider that there are different options to compute rest break effects and to take this into account (if necessary, in a more qualitative way by discussing it). Overall, this work is very useful contribution of the field, and I would invite you preparing a revision of your work that addresses all points together with a cover letter that provides point-by-point replies. 

We look forward to receiving your revised manuscript.

Kind regards,

Michael B. Steinborn, PhD

Section Editor

PLOS ONE

Journal Requirements:

"The work of Andrei Rusu was supported by a grant of the Romanian Ministry of Education and Research, CNCS - UEFISCDI, project number PN-III-P1-1.1-TE-2019-2032, within PNCDI III."

We note that you have provided funding information. However, funding information should not appear in the Acknowledgments section or other areas of your manuscript. We will only publish funding information present in the Funding Statement section of the online submission form. 

"The work of A. R. was supported by a grant of the Romanian Ministry of Education and Research, CNCS - UEFISCDI, project number PN-III-P1-1.1-TE-2019-2032, within PNCDI III."

3. We note that this manuscript is a systematic review or meta-analysis; our author guidelines therefore require that you use PRISMA guidance to help improve reporting quality of this type of study. Please upload copies of the completed PRISMA checklist as Supporting Information with a file name “PRISMA checklist”.

Reviewers' comments:

Reviewer's Responses to Questions

**Comments to the Author**

1. Is the manuscript technically sound, and do the data support the conclusions?

Reviewer #1: Yes

Reviewer #2: Partly

2. Has the statistical analysis been performed appropriately and rigorously? 

Reviewer #1: Yes

Reviewer #2: N/A

3. Have the authors made all data underlying the findings in their manuscript fully available?

Reviewer #1: Yes

Reviewer #2: Yes

4. Is the manuscript presented in an intelligible fashion and written in standard English?

Reviewer #1: Yes

Reviewer #2: Yes

5. Review Comments to the Author

Reviewer #1: Dear colleagues,

this is a very thorough meta-analysis on a very important topic from work and organizational research:

the effects of short rest breaks on strain, motivation and perfromance.

I really liked to read the paper and the quality, so far, is very impressive.

However, I have some further suggestions for improvement before a publication is warranted.

1) When reporting I2 please use superscript for '2' => I²

2) p.2, l.34: "Sub-groups analyses on performance types revealed significant effects only for tasks with less cognitive demands".

3) Overall, the data support the role of micro-breaks for well-being, while for performance, recovering from highly depleting tasks may need more than 10-minute breaks. Therefore, future studies should focus on this issue.

[Or what do you mean specifcally with 'process'?)

4) p. 5, l. 120: Please revise the sentence and refer to the study: for example

"For instance, in one study many of emplyoees reported break activities were negatively associated with increased energy (i.e., vitality) but positively related to fatigue [5]."

5) p.13., l. 303: "The main research questions were addressed using random-effects meta-analyses

based on Borenstein et al.'s [72] framework."

6) 13, l. 306: "two-sided p-values".

7) p.14. l329-330: τ² (Suberscript)

8) p.16: 365-376: The numbers do not fully match with the data in the flow diagramm: please check this

9) I missed a number of studies that might have matched with your inclusion criteria. Did you found them during search?

Singh, U., Ghadiri, A., Weimar, D., & Prinz, J. (2020). “Let’s have a break”: An experimental comparison of work-break interventions and their impact on performance. Journal of Business Research, 112, 128-135.

Blasche, G., Szabo, B., Wagner‐Menghin, M., Ekmekcioglu, C., & Gollner, E. (2018). Comparison of rest‐break interventions during a mentally demanding task. Stress and Health, 34(5), 629-638.

10.) p.21, l.385+386 should be deleted

11) p.33, l.665 "However, at least for well-being, the answer seems to be "any type of decoupling activity" (delete comma)

12) p.30: l.581: you write voluntary but to my impression the breaks were forced (involuntary)

13) p.30, l.587 fits with that meta-analysis

Sio, U. N., & Ormerod, T. C. (2009). Does incubation enhance problem solving? A meta-analytic review. Psychological Bulletin, 135(1), 94.

14) p.33: Another option would be adjusting for reliabilty of measure within the meta-analysis (see Schmidt-Hunter-Approach)

Reviewer #2: The authors present an manuscript that deals with an important topic, the influence of rest breaks on feelings and performance, which is well written overall and personally interesting to me as a reader. Being a professor in personnel psychology and statistics, I will judge the manuscript with a focus on concept and methodology. See below for detailed comments.

## the abstract need revision. the abstract should include the research question, the methods and the results but in a comprehensible way. statistics or other details or specifics should not be referred to in the abstract.

## The writing is good, and the manuscript is generally interesting to read. It is also theoretically well argued, and to the point. There are some points to consider. The first concerns the instruments to measures effects of breaks, it can be a rating scale or a performance output but the effect size is difficult to compare directly, even between performance outputs of different tasks, not to speak of between feelings and performance. It depends on how good the psychometric quality of the measure is. For example, 30 min of a vigilance task deliver only a few measurement units while 10 min of a self-paced task provide a huge amount of trials to obtain reliable measures, and statistics crucially depend on it. (suggested reference: Schumann et al. (2022). Restoration of attention by rest in a multitasking world: Theory, methodology, and empirical evidence. Frontiers in Psychology, 13, 867978. doi:10.3389/fpsyg.2022.867978

## definitions of micro breaks fit with the meaning used in work psychology. I have no problem with this definition because the authors explain everything well in the manuscript, so there is no ambivalence. On the other hand, researcher with a more cognitive experimental focus often differ in the use of what is a micro break versus a short break versus long breaks. This might be discussed.

## glucose and depletion of resources. in the manuscript, the authors often refer to how task deplete mental resource in a way that gives the impression of a strong underlying basis of biology and physiology. On one occasion, for example, glucose is referred to as a causal factor determining resources. This is a difficult question, I would suggest reconsidering this point, it is in my opinion not clear.

## tables. If possible, the tables should be presented in APA norm and maybe in a more economic way, The tables are huge and exceed a page. If the authors find ways to organise this more economically, good, otherwise, it is also okay, I would be fine with the present version.

6. PLOS authors have the option to publish the peer review history of their article (what does this mean?). If published, this will include your full peer review and any attached files.

Reviewer #1: **Yes: **Johannes Wendsche

Reviewer #2: No

---

## [Author Response · Author response to Decision Letter 0]

14 Jun 2022

Ref: PONE-D-22-07400

Title: Let's take a break! A systematic review and meta-analysis on the efficacy of micro-breaks for increasing well-being and performance

Authors: Patricia Albulescu, Irina Macsinga, Andrei Rusu, Coralia Sulea, Alexandra Bodnaru, Bogdan Tudor Tulbure

RESPONSE LETTER

Dear dr. Steinborn,

Thank you for granting us the opportunity to revise and resubmit our manuscript and thank you for the feedback and extremely valuable comments for improving the quality of the paper. In the following, we will answer each and every suggestion in a point-by-point manner.

But first of all, we want to mention that when we decided on the title for the manuscript, unfortunately, we did not pay enough attention to notice that there are other published articles (even in our search results) using a similar syntagm, such we used (i.e., "Let’s have a break"), in their title (e.g., Singh, U., Ghadiri, A., Weimar, D., & Prinz, J. (2020). “Let’s take a break”: An experimental comparison of work-break interventions and their impact on performance. Journal of Business Research, 112, 128-135.).

Thus, we suggest a slight modification to the tile, by using a syntagm that seems not to be found in other article titles (at least based on a google scholar search).

Initial title: Let's take a break! A systematic review and meta-analysis on the efficacy of micro-breaks for increasing well-being and performance

New title: "Give me a break!" A systematic review and meta-analysis on the efficacy of micro-breaks for increasing well-being and performance

Responses to the section editor’s observations

The main point and overall impression:

As you can see, both referees seem to consider you work important overall, while at the same time provided a whole number of additional points that might be useful to consider during the revision of the manuscript. To name the most important points, both referees raised the methodical points of how to best quantify study results, and when to consider a qualitative evaluation of aspects of the studies. Here, both refer to findings on the connection between experience and performance (which sometimes go together, other times not), and, to the reliability of the underlying tests and performance measures. My personal suggestion is to also consider that there are different options to compute rest break effects and to take this into account (if necessary, in a more qualitative way by discussing it). 

Overall, this work is very useful contribution of the field, and I would invite you preparing a revision of your work that addresses all points together with a cover letter that provides point-by-point replies.

<Response> Thank you for your positive feedback and the chance to revise the manuscript! As we also developed in the responses to each reviewer, the possibility to control for psychometric artefacts (reliability estimates) based on Hunter and Schmidt’s approach was limited by inconsistencies in reporting such estimates for vigor and fatigue, and with an almost total lack of reporting for the performance outcomes (with two exceptions). Hence the decision to entirely apply Borenstein et al.’s framework for the meta-analysis. But we agree that this may be an important aspect, and as you rightfully suggested, we developed and inserted in the manuscript a detailed discussion of this limitation and its implications. We also discussed and accentuated the cautionary note regarding the diversity and lack of consistency when it comes to approaches for performance and the consequential potential sources of bias. These aspects were developed (1) at the end of the Introduction (by adding new details to the already existing mentions on measurement and conceptualization issues), (2) in the Description of the included studies subsection from Results (we developed on the operationalizations being used in the included studies by critically uprising their reliability, threats to reliability and the general inconsistencies in approaches), (3) in the Discussion when presenting the results on the performance outcome as also in the Limitations subsection. This way the reader will get a stronger critical appraisal of this body of literature.

Additional requirements:

<Response> The elements related to style requirements were checked again against PlosOne’s specifications.

2. We note that you have provided funding information. However, funding information should not appear in the Acknowledgments section or other areas of your manuscript. We will only publish funding information present in the Funding Statement section of the online submission form. 

Please remove any funding-related text from the manuscript and let us know how you would like to update your Funding Statement.

<Response> The “Financial Disclosure Statement” was removed from the manuscript. The information related to funding can be found in the “Funding Statement” document. No update is required on this funding statement.

3. We note that this manuscript is a systematic review or meta-analysis; our author guidelines therefore require that you use PRISMA guidance to help improve reporting quality of this type of study. Please upload copies of the completed PRISMA checklist as Supporting Information with a file name “PRISMA checklist”.

<Response> The filled-in PRISMA checklist was uploaded as Supporting Information the following journal guidelines (i.e., Supporting Information files should be saved as “S1_Fig.tif”, “S1_File.pdf”, etc.) as “S4_Table”. After revision, the file name was changed from “S4_Table” to “S4_Table. PRISMA Checklist”.

Responses to the first reviewer’s comments

Overall impression: 

This is a very thorough meta-analysis on a very important topic from work and organizational research: the effects of short rest breaks on strain, motivation and perfromance. I really liked to read the paper and the quality, so far, is very impressive. However, I have some further suggestions for improvement before a publication is warranted.

<Response> Thank you for your positive feedback and appreciation!

Comments #1 to #7: 

(1) When reporting I2 please use superscript for '2' => I²; 

(2) p.2, l.34: "Sub-groups analyses on performance types revealed significant effects only for tasks with less cognitive demands". 

(3) Overall, the data support the role of micro-breaks for well-being, while for performance, recovering from highly depleting tasks may need more than 10-minute breaks. Therefore, future studies should focus on this issue. [Or what do you mean specifcally with 'process'?) 

(4) p. 5, l. 120: Please revise the sentence and refer to the study: for example

"For instance, in one study many of emplyoees reported break activities were negatively associated with increased energy (i.e., vitality) but positively related to fatigue [5]."

 (5) p.13., l. 303: "The main research questions were addressed using random-effects meta-analyses based on Borenstein et al.'s [72] framework."

(6) 13, l. 306: "two-sided p-values". 

(7) p.14. l329-330: τ² (Suberscript)

<Response> Thank you for the detailed observations! All these suggestions were considered and can be found modified (with track changes) in the revised manuscript.

Comment #8: p.16: 365-376: The numbers do not fully match with the data in the flow diagram: please check this.

<Response> Thank you for noticing this! We checked the information and the results of the search process. By mistake, in the text were reported 15 records instead of 25 excluded with reason (as correctly reported on the flow chart). The correction was made accordingly.

Comment #9: I missed a number of studies that might have matched with your inclusion criteria. Did you find them during search?

Singh, U., Ghadiri, A., Weimar, D., & Prinz, J. (2020). “Let’s have a break”: An experimental comparison of work-break interventions and their impact on performance. Journal of Business Research, 112, 128-135. 

Blasche, G., Szabo, B., Wagner‐Menghin, M., Ekmekcioglu, C., & Gollner, E. (2018). Comparison of rest‐break interventions during a mentally demanding task. Stress and Health, 34(5), 629-638.

<Response> We found one of the suggested studies. The first study, conducted by Singh et al. (2020), was found during the systematic search but was excluded from further analysis (records excluded with reason) as it was deemed not eligible because it tested three parallel interventions with no control group to serve as a comparison (see the Comparator criterion from the PICOS – ‘Eligibility criteria’ section of the manuscript). The second study done by Blasche et al. (2018) did not appear in the initial systematic literature search. Analyzing it and applying our inclusion/exclusion criteria, we found it also not eligible as the task in which participants were engaged (from which the micro-break was taken) does not have a correspondence in the occupational setting. Moreover, as the authors conclude, “Though an academic class imposes demands also found in other kinds of mental work, and both academic classwork and office work lead to fatigue, office work and work in an academic class are obviously not identical. This warrants caution in generalizing the current findings to other contexts of mental work (p. 636)”. All other papers included in this meta-analysis used tasks with some degree of similarity with those specific to the job environment.

However, after reading again our manuscript, we noticed that the exclusion of the non-work-related studies was not made clear enough, thus in the revised version, we made this explicit mention for the eligibility criteria. Moreover, based on Blasche et al.’s (2018) suggestion we also lowered our tone on a cautionary note in the discussion of the implications of our meta-analysis for the educational setting.

Comments #10 to #12: 

(10) p.21, l.385+386 should be deleted 

(11) p.33, l.665 "However, at least for well-being, the answer seems to be "any type of decoupling activity" (delete comma)

(12) p.30: l.581: you write voluntary but to my impression the breaks were forced (involuntary)

<Response> All suggestions were included in the new version of the manuscript. 

Comment #13: p.30, l.587 fits with that meta-analysis

Sio, U. N., & Ormerod, T. C. (2009). Does incubation enhance problem solving? A meta-analytic review. Psychological Bulletin, 135(1), 94.

<Response> We checked the suggested reference and thank you for pointing it out. Indeed, it fits with the drawn conclusion, and we included a mention based on it in the new version of the manuscript. Accordingly, the References section was also updated. 

Comment #14: p.33: Another option would be adjusting for reliabilty of measure within the meta-analysis (see Schmidt-Hunter-Approach)

<Response> When we started planning the meta-analysis, we also discussed the possibility to control for psychometric artefacts based on Hunter and Schmidt’s approach. However, while screening the full-text articles and prior to beginning the coding process, we realized that there were large inconsistencies in reporting the reliabilities of the outcomes of interests. Only for some of the vigor and fatigue measures, the authors reported reliabilities (some were also one-item instruments). Moreover, for what we referred to as 'objective measures' of performance (e.g., error rates, reaction times, etc.), there was no such information. Hence, we decided to entirely apply Borenstein et al.’s framework for the meta-analysis. But we agree that this may be an important aspect, and as also per the Editor’s suggestion, we inserted in the manuscript a description and a discussion of this limitation and its implications.

Responses to the second reviewer’s comments

Overall impression: 

The authors present an manuscript that deals with an important topic, the influence of rest breaks on feelings and performance, which is well written overall and personally interesting to me as a reader. Being a professor in personnel psychology and statistics, I will judge the manuscript with a focus on concept and methodology. See below for detailed comments.

<Response> Thank you for your positive feedback and appreciation!

Comment #1: the abstract needs revision. the abstract should include the research question, the methods and the results but in a comprehensible way. statistics or other details or specifics should not be referred to in the abstract.

<Response> The statistical details were removed from the abstract. We also reformulated the aims and methods details from the abstract in a more comprehensive way.

Comment #2: The writing is good, and the manuscript is generally interesting to read. It is also theoretically well argued, and to the point. There are some points to consider. The first concerns the instruments to measures effects of breaks, it can be a rating scale or a performance output but the effect size is difficult to compare directly, even between performance outputs of different tasks, not to speak of between feelings and performance. It depends on how good the psychometric quality of the measure is. For example, 30 min of a vigilance task deliver only a few measurement units while 10 min of a self-paced task provide a huge amount of trials to obtain reliable measures, and statistics crucially depend on it. (suggested reference: Schumann et al. (2022). Restoration of attention by rest in a multitasking world: Theory, methodology, and empirical evidence. Frontiers in Psychology, 13, 867978. doi:10.3389/fpsyg.2022.867978

<Response> Thank you for stressing this aspect! Indeed, it represents an important limitation of the current literature. As also the Editor suggested, we discussed it in several points of our manuscript. Moreover, thanks to the suggested review (Schumann et al., 2022), we also referred to some of their valuable critical insights. Precisely, (1) we began by further developing the measurement and conceptualization issues already mentioned towards the end of the Introduction, (2) continued by describing the state of the reliabilities of the used measures in the description of the sample of studies from the Results section, and (3) finally, we also addressed this issue in the Discussion by stressing out, among others, that "there are multiple methodological factors (e.g., employment of arbitrary tasks and/or unreliable or unaudited performance measures; insufficient number of trials) which could result in artefactual variance", and "even if taking into account their purpose, all these measures can be grouped under the umbrella term of performance, the many conceptual and operational differences between them could also make one rightfully argue against their aggregation. Hence, these particular results should be taken with increased caution.", as we also suggest " future studies might test the conditions under which respites positively affect subsequent performance by applying more standardized experimental paradigms and increasing consensus on methodological approaches." (These are just some examples of new mentions that we added to the manuscript.)

Comment #3: definitions of micro breaks fit with the meaning used in work psychology. I have no problem with this definition because the authors explain everything well in the manuscript, so there is no ambivalence. On the other hand, researcher with a more cognitive experimental focus often differ in the use of what is a micro break versus a short break versus long breaks. This might be discussed.

<Response> We searched in the cognitive-focused literature, and we didn’t find a clear differentiation between micro-breaks, short-breaks, and long breaks. We have cited several studies in cognitive psychology that emphasize the importance of refocusing attention after the break and we have highlighted a link between the three attention networks and breaks. We have inserted the new paragraph in the text. 

Comment #4: glucose and depletion of resources. in the manuscript, the authors often refer to how task deplete mental resource in a way that gives the impression of a strong underlying basis of biology and physiology. On one occasion, for example, glucose is referred to as a causal factor determining resources. This is a difficult question, I would suggest reconsidering this point, it is in my opinion not clear.

<Response> In order to not create confusion related to the biological underlying mechanism, we deleted the entire phrase and the references.

Comment #5: tables. If possible, the tables should be presented in APA norm and maybe in a more economic way, The tables are huge and exceed a page. If the authors find ways to organise this more economically, good, otherwise, it is also okay, I would be fine with the present version.

<Response> All the included tables meet PlosOne’s style formatting requirements, which do not follow APA norms. Unfortunately, we did not find a more economical way to organize the information.

---

## [Decision Letter · Decision Letter 1]

27 Jun 2022

PONE-D-22-07400R1"Give me a break!" A systematic review and meta-analysis on the efficacy of micro-breaks for increasing well-being and performancePLOS ONE

Dear Dr. Macsinga,

Thank you for submitting your manuscript to PLOS ONE. After careful consideration, we feel that it has merit but does not fully meet PLOS ONE’s publication criteria as it currently stands. Therefore, we invite you to submit a revised version of the manuscript that addresses the points raised during the review process. Editor comment. The manuscript seem now ready for publication. I would accept your manuscript for publication after considering some minor points provided by reviewer 1. There will be no further rounds, which means the manuscript will officially be accepted after resubmission and editorial check of the final version. Please submit your revised manuscript by Aug 11 2022 11:59PM. If you will need more time than this to complete your revisions, please reply to this message or contact the journal office at plosone@plos.org. Please include the following items when submitting your revised manuscript:A rebuttal letter that responds to each point raised by the academic editor and reviewer(s). You should upload this letter as a separate file labeled 'Response to Reviewers'.A marked-up copy of your manuscript that highlights changes made to the original version. You should upload this as a separate file labeled 'Revised Manuscript with Track Changes'.An unmarked version of your revised paper without tracked changes. You should upload this as a separate file labeled 'Manuscript'.If applicable, we recommend that you deposit your laboratory protocols in protocols.io to enhance the reproducibility of your results. Protocols.io assigns your protocol its own identifier (DOI) so that it can be cited independently in the future. For instructions see: https://journals.plos.org/plosone/s/submission-guidelines#loc-laboratory-protocols. Additionally, PLOS ONE offers an option for publishing peer-reviewed Lab Protocol articles, which describe protocols hosted on protocols.io. Read more information on sharing protocols at https://plos.org/protocols?utm_medium=editorial-email&utm_source=authorletters&utm_campaign=protocols.

We look forward to receiving your revised manuscript.

Kind regards,

Michael B. Steinborn, PhD

Section Editor

PLOS ONE

Journal Requirements:

Reviewers' comments:

Reviewer's Responses to Questions

**Comments to the Author**

1. If the authors have adequately addressed your comments raised in a previous round of review and you feel that this manuscript is now acceptable for publication, you may indicate that here to bypass the “Comments to the Author” section, enter your conflict of interest statement in the “Confidential to Editor” section, and submit your "Accept" recommendation.

Reviewer #1: All comments have been addressed

Reviewer #2: All comments have been addressed

2. Is the manuscript technically sound, and do the data support the conclusions?

Reviewer #1: Yes

Reviewer #2: Yes

3. Has the statistical analysis been performed appropriately and rigorously? 

Reviewer #1: Yes

Reviewer #2: Yes

4. Have the authors made all data underlying the findings in their manuscript fully available?

Reviewer #1: Yes

Reviewer #2: (No Response)

5. Is the manuscript presented in an intelligible fashion and written in standard English?

Reviewer #1: Yes

Reviewer #2: Yes

6. Review Comments to the Author

Reviewer #1: I would like to thank the authors for revising the manuscript in line with my earlier comments.

I have some few (minor) recommendation for further improvement before a publication is warranted.

1) Abstract

Please inlcude specific information on the final sample size (number of studies, number of participants; i.e., data from 19 publications with 22 independent study samples, N = ...; data from p. 16) and in addition the

average effect size estimates of microbreaks. This information will be helfpul for readers to get the most important results

from your review in a quick way. I do agree with reviewer 2 that not all statistical information (95%CIs, p values, I²) is necessary in the abstract. But d-values and corresponding ks are 'must-haves' from my perspective.

2) p.16, l.380: do you mean 22 independent study samples from 19 publications?

3) Figure with PRIMSA scheme: Eligibility

Full text assessed = 43 articles

then you exclude 24

=> difference is 19

but you report 22 articles for the qualitative analysis ???

In addition, number of publications shlould be marked with k (n is usally used for participant sample size).

In the final step please report:

Included: k=19 publications with 22 independent study samples

(correct?)

4) p. 30, l. 575: Our results reveal that micro-break...

5) p.24, l.469: A d = 13.43 is extreme from my experience of this literature.

Have you checked effect coding and effect calculations from this specific study,

maybe by different coders?

6) According to your registered study protocol only control-group design studies could be included.

However, during your literature search you find that there are many studies using

no control group /within-subject designs.

I agree that that this design might affect the valdity of results.

On the other hand, even the design and measures of the included studies are largely heterogenous.

So at least for the small effects sizes for performance outcomes it might be interesting to integrate and discuss some of the

relevant excluded study results in the discussion. Do these studies found some performance effects?

For instance, consider tthi studies:

Engelmann, C., Schneider, M., Kirschbaum, C., Grote, G., Dingemann, J., Schoof, S., & Ure, B. M. (2011). Effects of intraoperative breaks on mental and somatic operator fatigue: a randomized clinical trial. Surgical endoscopy, 25(4), 1245–1250. https://doi.org/10.1007/s00464-010-1350-1

Engelmann, C., Schneider, M., Grote, G., Kirschbaum, C., Dingemann, J., Osthaus, A., & Ure, B. (2012). Work breaks during minimally invasive surgery in children: patient benefits and surgeon's perceptions. European journal of pediatric surgery : official journal of Austrian Association of Pediatric Surgery ... [et al] = Zeitschrift fur Kinderchirurgie, 22(6), 439–444. https://doi.org/10.1055/s-0032-1322542

Park, A. E., Zahiri, H. R., Hallbeck, M. S., Augenstein, V., Sutton, E., Yu, D., Lowndes, B. R., & Bingener, J. (2017). Intraoperative "Micro Breaks" With Targeted Stretching Enhance Surgeon Physical Function and Mental Focus: A Multicenter Cohort Study. Annals of surgery, 265(2), 340–346. https://doi.org/10.1097/SLA.0000000000001665

It is hard to conduct real intervention-control-group designs in such work settings. For instance, consider the Engelmann studies that showed real performance effects in a way that breaks improved patient outcomes (not only the concentration of the surgeons).

However, such studies found positive effects of breaks on performance which bolster the argument that breaks might even improve perfomance. Another argument is also that in many rest break studies the intervention group with additional rest breaks has actually a shorter total time on task (or work duration). Thus, significant increases in work performance (if this is assessed) are not really to be expected, it will be fine if this might be similar to the control group which has a longer time on task.

In sum, I will prefer if you could add some additional insights relating to performance effects from the excluded study types into the discussion.

Reviewer #2: The manuscript has improved greatly, and I would recommend this work for publication. I have no further comments.

7. PLOS authors have the option to publish the peer review history of their article (what does this mean?). If published, this will include your full peer review and any attached files.

Reviewer #1: No

Reviewer #2: No

---

## [Author Response · Author response to Decision Letter 1]

19 Jul 2022

Responses to the first reviewer’s comments

Reviewer #1: I would like to thank the authors for revising the manuscript in line with my earlier comments. I have some few (minor) recommendation for further improvement before a publication is warranted.

<Response> Thank you for your feedback and suggestions for improvement of our manuscript. We approached every comment carefully and integrated the suggestions in the new manuscript. In the following, we also responded to each and every comment.

1) Abstract

Please inlcude specific information on the final sample size (number of studies, number of participants; i.e., data from 19 publications with 22 independent study samples, N = ...; data from p. 16) and in addition the average effect size estimates of microbreaks. This information will be helfpul for readers to get the most important results from your review in a quick way. I do agree with reviewer 2 that not all statistical information (95%CIs, p values, I²) is necessary in the abstract. But d-values and corresponding ks are 'must-haves' from my perspective.

<Response> Thank you for this suggestion. We included in the abstract a phrase summarizing the retrieved number of studies, as well as the effect sizes for each of the outcomes of interest.

2) p.16, l.380: do you mean 22 independent study samples from 19 publications?

<Response> Thank you for pointing this mistake out. The observation is correct, and we made a change here, at line 382 using the phrase “22 independent study samples” instead of the initial one, reading “22 studies”, as this, we hope, will be a clearer message. 

3) Figure with PRIMSA scheme: Eligibility

Full text assessed = 43 articles

then you exclude 24

=> difference is 19

but you report 22 articles for the qualitative analysis ???

In addition, number of publications shlould be marked with k (n is usally used for participant sample size).

In the final step please report:

Included: k=19 publications with 22 independent study samples

(correct?)

<Response> Thank you for pointing this out! The changes in the Figure 1 were processed as follows: 

1. Changed n with k in the entire PRISMA flow diagram

2. Articles included in the qualitative synthesis (k = 19) 

3. Studies included in quantitative synthesis (meta-analysis) (k = 19 publications with 22 independent study samples)

4) p. 30, l. 575: Our results reveal that micro-break...

<Response> Thank you. We implemented the suggestion which can be found at line 577, page 30.

5) p.24, l.469: A d = 13.43 is extreme from my experience of this literature.

Have you checked effect coding and effect calculations from this specific study, maybe by different coders?

<Response> For all coding we used two independent raters. Also, all the calculations were automatically done by the software we used (Comprehensive Meta-Analysis v 3.0), based on means, standard deviations, and sample sizes as input data. However, this particular effect also puzzled us and made us check it meticulously (since human error may hay have slipped in extracting the descriptive results from the manuscript). After inspecting the data against the effect size formula, we realized that the standard deviations of each mean are very low, hence the very large d value (after the standardization of the mean difference). Bellow, we will illustrate the calculus:

Experimental group data: Mean = 2.14, Std. Dev. = 0.12, Sample size = 51

Control group data: Mean = 3.72, Std. Dev. = 0.11, Sample size = 17

d = |(M experimental – M control)| / Pooled SD

d = |(2.14 – 3.72)| / 0.118

d = 1.58 / 0.118

d = 13.39

Some observations: (1) the obtained result varies slightly from the one outputted by Comprehensive Meta-Analysis (we do not know the exact formulas behind the software, maybe include the correction for sample size or something similar); (2) the difference in means is computed in module since a lower score on fatigue (as is the case for the experimental condition compared to the control one) is the expected result; (3) the actual design of the study included 4 groups (2 experimental, the difference being the theme of the micro-break, nature exposure <n =26=""> or urban exposure <n =="" 25="">, and 2 control, one with the same depleting task but without micro-break <n =="" 17="">, and another one without any manipulation <n =="" 19="">); since the later control fell outside our purpose (because we aimed at comparing groups who passed a similar depleting treatment, the only difference being the micro-break manipulation, as to draw conclusions on the effect of micro-breaks in working conditions), we pulled the data of the two experimental groups into one (n = 51; also these groups had similar results, i.e., means of 2.2 and 2.1) and compared it against the eligible control (n = 17). In their study, Beute and de Kort (2014), reported only the omnibus comparison between the four conditions (ANOVA analysis), without finding a significant overall effect (possible because of the small samples from each condition and the low and homogeneous variances), thus, they did not report further analyses on these data. Moreover, even for the significant differences, they reported only the overall effect size (partial etta squared) and no pairwise effects (hence the reason no similar effect size was reported in the manuscript).

6) According to your registered study protocol only control-group design studies could be included.

However, during your literature search you find that there are many studies using no control group /within-subject designs.

I agree that that this design might affect the valdity of results.

On the other hand, even the design and measures of the included studies are largely heterogenous.

So at least for the small effects sizes for performance outcomes it might be interesting to integrate and discuss some of the relevant excluded study results in the discussion. Do these studies found some performance effects?

For instance, consider tthi studies:

Engelmann, C., Schneider, M., Kirschbaum, C., Grote, G., Dingemann, J., Schoof, S., & Ure, B. M. (2011). Effects of intraoperative breaks on mental and somatic operator fatigue: a randomized clinical trial. Surgical endoscopy, 25(4), 1245–1250. https://doi.org/10.1007/s00464-010-1350-1

Engelmann, C., Schneider, M., Grote, G., Kirschbaum, C., Dingemann, J., Osthaus, A., & Ure, B. (2012). Work breaks during minimally invasive surgery in children: patient benefits and surgeon's perceptions. European journal of pediatric surgery : official journal of Austrian Association of Pediatric Surgery ... [et al] = Zeitschrift fur Kinderchirurgie, 22(6), 439–444. https://doi.org/10.1055/s-0032-1322542

Park, A. E., Zahiri, H. R., Hallbeck, M. S., Augenstein, V., Sutton, E., Yu, D., Lowndes, B. R., & Bingener, J. (2017). Intraoperative "Micro Breaks" With Targeted Stretching Enhance Surgeon Physical Function and Mental Focus: A Multicenter Cohort Study. Annals of surgery, 265(2), 340–346. https://doi.org/10.1097/SLA.0000000000001665

It is hard to conduct real intervention-control-group designs in such work settings. For instance, consider the Engelmann studies that showed real performance effects in a way that breaks improved patient outcomes (not only the concentration of the surgeons).

However, such studies found positive effects of breaks on performance which bolster the argument that breaks might even improve perfomance. Another argument is also that in many rest break studies the intervention group with additional rest breaks has actually a shorter total time on task (or work duration). Thus, significant increases in work performance (if this is assessed) are not really to be expected, it will be fine if this might be similar to the control group which has a longer time on task.

In sum, I will prefer if you could add some additional insights relating to performance effects from the excluded study types into the discussion.

<Response> Thank you for this suggestion! We further developed the discussion, looking at all the studies on performance from the 43 papers included in the eligibility assessing phase (including the ones that you suggested). Hence, we added a more extensive overview on the effect of breaks on performance (see Pages 33-34, lines 662-694). Because we included papers not cited previously throughout the manuscript, the in-text citations and reference list were also updated.

Responses to the second reviewer’s comments

Reviewer #2: The manuscript has improved greatly, and I would recommend this work for publication. I have no further comments.

<Response> Thank you for all the suggestions made during the first round of feedback, which made it possible to have this new and improved version of the manuscript. Also, thank you for your kind support during this review process.

---

## [Editor Report · Decision Letter 2]

20 Jul 2022

"Give me a break!" A systematic review and meta-analysis on the efficacy of micro-breaks for increasing well-being and performance

PONE-D-22-07400R2

Dear Dr. Macsinga,

the manuscript has even further improved and is ready for publication now. We’re pleased to inform you that your manuscript has been judged scientifically suitable for publication and will be formally accepted for publication once it meets all outstanding technical requirements.

Kind regards,

Michael B. Steinborn, PhD

Section Editor

PLOS ONE
---

## [Editor Report · Acceptance letter]

4 Aug 2022

PONE-D-22-07400R2 

"Give me a break!" A systematic review and meta-analysis on the efficacy of micro-breaks for increasing well-being and performance 

Dear Dr. Macsinga:

I'm pleased to inform you that your manuscript has been deemed suitable for publication in PLOS ONE. Congratulations! Your manuscript is now with our production department. 

Kind regards, 

on behalf of

Dr. Michael B. Steinborn 

Section Editor

PLOS ONE